# Postprandial FGF19-induced phosphorylation by Src is critical for FXR function in bile acid homeostasis

Sangwon Byun [1], Dong-Hyun Kim[1], Daniel Ryerson[1], Young-Chae Kim [1], Hao Sun[1], Bo Kong [2], Peter Yau[3], Grace Guo[2], H. Eric Xu[4], Byron Kemper[1] & Jongsook Kim Kemper[1]

Farnesoid-X-Receptor (FXR) plays a central role in maintaining bile acid (BA) homeostasis by transcriptional control of numerous enterohepatic genes, including intestinal FGF19, a hormone that strongly represses hepatic BA synthesis. How activation of the FGF19 receptor at the membrane is transmitted to the nucleus for transcriptional regulation of BA levels and whether FGF19 signaling posttranslationally modulates FXR function remain largely unknown. Here we show that FXR is phosphorylated at Y67 by non-receptor tyrosine kinase, Src, in response to postprandial FGF19, which is critical for its nuclear localization and transcriptional regulation of BA levels. Liver-specific expression of phospho-defective Y67F-FXR or Src downregulation in mice results in impaired homeostatic responses to acute BA feeding, and exacerbates cholestatic pathologies upon drug-induced hepatobiliary insults. Also, the hepatic FGF19-Src-FXR pathway is defective in primary biliary cirrhosis (PBC) patients. This study identifies Src-mediated FXR phosphorylation as a potential therapeutic target and biomarker for BA-related enterohepatic diseases.

[1] Department of Molecular and Integrative Physiology, University of Illinois at Urbana-Champaign, Urbana, 61801 IL, USA. [2] Department of Pharmacology and Toxicology, Ernest Mario School of Pharmacy, Rutgers University, Piscataway, 08854 NJ, USA. [3] Proteomics Center, University of Illinois at Urbana-Champaign, Urbana, 61801 IL, USA. [4] Laboratory of Structure Sciences, Van Andel Research Institute, Grand Rapids, 49503 MI, USA. These authors contributed equally: Sangwon Byun, Dong-Hyun Kim, Daniel Ryerson. Correspondence and requests for materials should be addressed to J.K.K. (email: jongsook@illinois.edu)

Bile acids (BAs) are amphipathic steroid molecules that aid in digestion of dietary lipids, but also function as signaling molecules that profoundly impact metabolism and energy balance by activating the nuclear receptor, farnesoid X receptor (FXR, NR1H4), and a membrane G-protein-coupled receptor, TGR5[1–5]. Due to their detergent-like nature, excess levels of BAs are toxic and lead to cellular injury, which can further progress to fatal diseases, such as liver fibrosis and cirrhosis, and liver and intestinal cancer[6,7]. FXR is the primary BA biosensor and plays a central role in maintaining BA homeostasis and protecting from BA-induced toxicity[8,9].

In response to a meal, FXR is activated by transiently elevated physiological concentrations of BAs in enterohepatic tissues and mediates postprandial regulation of BA, cholesterol, lipid, and glucose metabolism[2,10] and inhibition of autophagy-mediated lipid catabolism[11,12] to maintain homeostasis. FXR transcriptionally regulates numerous genes involved in nearly every aspect of BA metabolism, including synthesis, transport, and recycling. BA-activated FXR represses hepatic BA synthesis by inducing expression of two key regulators, an intestinal hormone, FGF19 (human FGF19, mouse FGF15), and an orphan nuclear receptor, SHP[9,10]. FXR also regulates BA levels through induction of other target genes, including MAFG, a global transcriptional repressor of BA synthesis[13], LSD1, a key repressive epigenetic component in a SHP complex[14,15], and β-Klotho (βKL), the obligate co-receptor for FGF19[16].

The physiological role of the FXR-FGF19 gut-liver pathway in feedback repression of hepatic BA synthesis has been established[17–19]. After a meal, BA-activated FXR induces expression of FGF19 in the ileum, and the secreted FGF19 binds to and activates a hepatic cell surface receptor, FGF receptor 4 (FGFR4), and its co-receptor, βKL, in the late fed-state[20,21]. Binding of FGF19 to the receptor complex triggers intracellular signaling pathways that mediate feedback repression of hepatic BA synthesis, and also other postprandial responses, including stimulation of glycogen and protein synthesis and suppression of autophagy-mediated lipid catabolism, independent of insulin action[15,20,22]. The in vivo functional role of the FGF15/19 signaling in BA regulation has been confirmed in genetic mice lacking FGF15, FGFR4, or βKL[20,23,24]. Despite the discovery of a pivotal role of FGF19 in regulating BA levels, it remains poorly understood how the activation of the FGF19 receptor at the membrane is relayed into the nucleus for transcriptional control of BA-regulating genes.

Nuclear receptors are ligand-regulated transcriptional factors that play crucial roles in mammalian physiology and disease[8]. Transcriptional activity of nuclear receptors is primarily regulated by lipophilic ligands, but their functions are also significantly modulated by cellular signaling-induced post-translational modifications (PTMs), which can alter the overall response to their ligands[25]. PTMs of nuclear receptors, including FXR, SHP, and LRH-1, play an important role in transcriptional control of metabolism in animals in vivo, intriguingly in a gene-specific manner, and dysregulation of these PTMs is associated with metabolic disorders[26–30]. While FXR induces the expression of FGF15/19, it remains unknown whether transcriptional activity of FXR can be physiologically modulated in a feedback manner by FGF19 signal-induced PTMs.

Utilizing proteomics, genomics, metabolomics analyses, as well as, metabolic analyses, we now show that FXR is phosphorylated at Y67 by non-receptor tyrosine kinase Src upon postprandial FGF19 signaling in hepatocytes, and that the FGF19-induced Y67-FXR phosphorylation is critical for its transcriptional regulation of BA levels. Finally, we present intriguing data that the hepatic FGF19-Src-FXR phosphorylation signaling axis is likely defective in primary biliary cirrhosis (PBC, also known as primary biliary cholangitis) patients.

## Results

### Y67 in FXR is the primary Tyr-phosphorylation site after FGF19 treatment

To determine whether FXR is a target of FGF19-induced phosphorylation, primary mouse hepatocytes (PMH) from FXR-KO mice, exogenously expressing flag-FXR, were treated with FGF19, and the FGF19-induced phosphorylation site(s) were identified by tandem mass spectrometry analysis. The LC-MS/MS analysis identified three Tyr (Y) phosphorylated residues, Y46, Y49, and Y67, in FXR (Supplementary Fig. 1a). Mutation of Y67 to Phe nearly completely abolished phosphorylation at Tyr of flag-FXR, while mutation of Y46 or Y49 did not reduce the phosphorylation (Fig. 1a), suggesting that Y67 is the primary Tyr phosphorylation site. Notably, Y67 is the most highly predicted Tyr phosphorylation site in FXR by bioinformatics analysis with NetPhos 3.1 (Supplementary Fig. 1b).

Consistent with these findings, FGF19 treatment led to transient increases in phosphorylated (p)-Tyr-FXR levels that peaked at 10 min and returned to basal levels after 30–60 min (Fig. 1b) while treatment of hepatocytes with insulin or a synthetic FXR agonist, GW4064, did not significantly increase Tyr-FXR phosphorylation (Fig. 1c). Further, feeding mice with chow supplemented with an FXR agonist, cholic acid (CA), for 3 h, which should lead to FXR induction of endogenous FGF15, dramatically increased p-Tyr-FXR levels in mouse liver extracts (Fig. 1d).

To detect phosphorylation of endogenous FXR in mouse liver at Y67, an FXR antibody specific for Y67 phosphorylation was developed (Supplementary Fig. 1c). Feeding mice with CA chow or treatment with GW4064, both of which induce intestinal expression of Fgf15, or FGF19 treatment increased p-Y67-FXR levels in C57BL/6 mice, but not in FXR-KO mice (Fig. 1e). Notably, the Y67 residue in FXR is highly conserved among vertebrates (Supplementary Fig. 1d), suggesting the functional importance of this residue. These results indicate that FXR is phosphorylated at Y67 in response to FGF19.

### FXR is phosphorylated at Y67 in response to feeding

After a meal, FGF15/19 levels rise[31] and mediate postprandial metabolic responses, including repression of BA synthesis[20,22]. To test whether FGF19-induced Y67-FXR phosphorylation was important for this physiological response, we determined the effects of feeding on Y67-FXR phosphorylation in control C57BL/6 and FGF15-KO mice. Feeding for 6 h after fasting markedly increased hepatic p-Y67-FXR levels in C57BL/6 mice, but not in FGF15-KO mice (Fig. 1f). In IHC studies, feeding increased p-Y67-FXR levels, particularly in the nucleus, while this increase was not detected in FGF15-KO mice (Fig. 1g, left). Further, total FXR levels in the nucleus were markedly decreased in FGF15-KO mice compared to control mice after feeding (Fig. 1g, right), suggesting that Y67-FXR phosphorylation may have a role in its nuclear localization. These results demonstrate that endogenous FXR is phosphorylated at Y67 in an FGF15-dependent manner during physiological feeding/fasting cycles.

### FXR-mediated regulation of hepatic BA-related genes is attenuated by the p-defective Y67F mutation

To investigate the hepatic function of Y67-FXR phosphorylation, FXR was specifically knocked out in the livers of FXR floxed mice by infection of AAV-TBG-Cre and either FXR-WT or p-defective Y67F-FXR mutant was expressed by co-injection of AAV-TBG viruses for each (Fig. 2a). Infection with AAV-TBG-Cre blocked expression FXR in liver but not in the ileum (Fig. 2b) as expected since Cre expression is driven by a hepatocyte-specific thyroxine-binding globulin (TBG) promoter. Protein levels of the expressed FXR or Y67F-FXR were similar to normal endogenous levels detected in

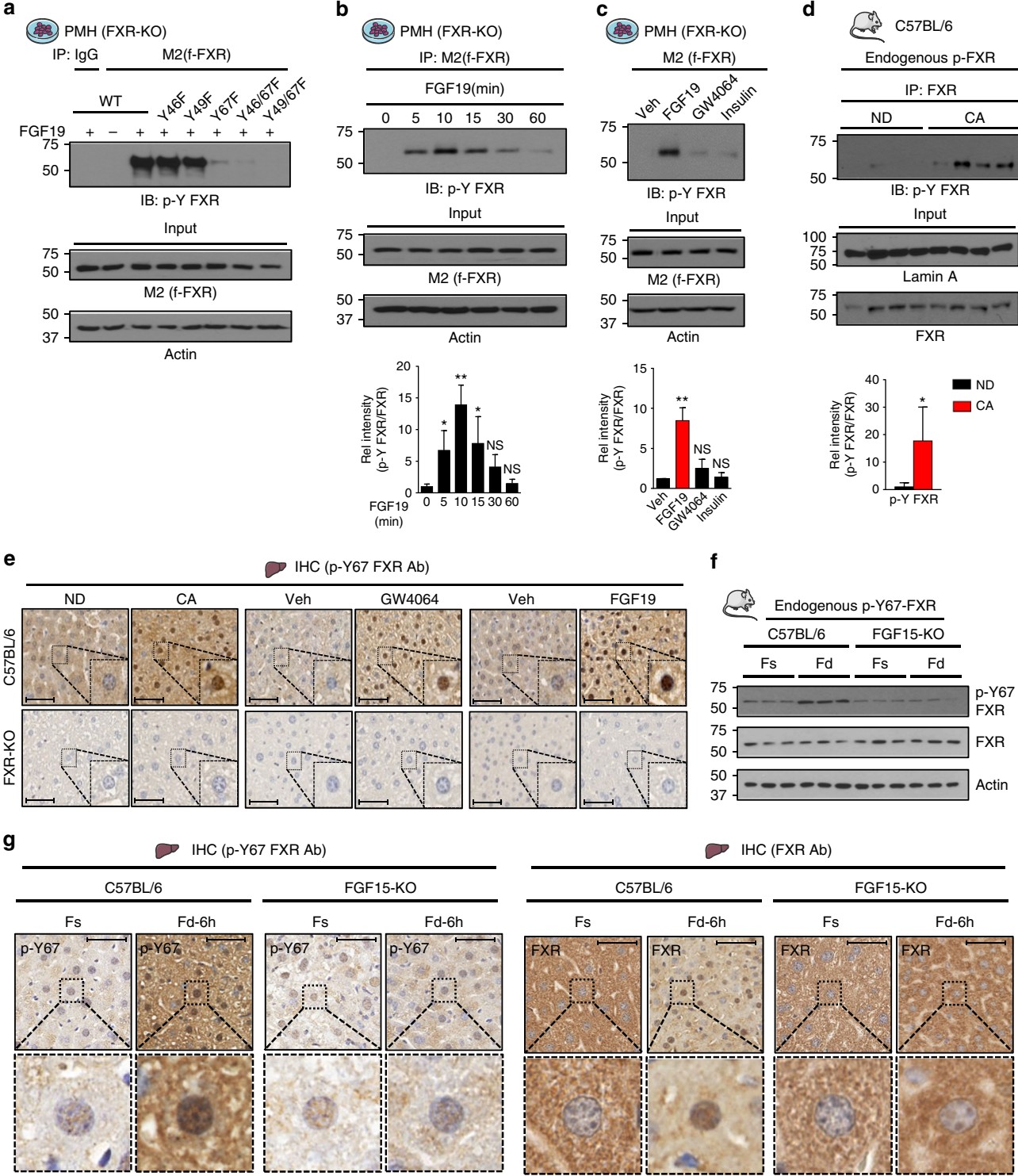

**Fig. 1** FXR is phosphorylated at Y67 by FGF19. **a** Flag-FXR-WT, or its Tyr-FXR mutants as indicated, was expressed in PMH from FXR-KO mice. Cells were treated with vehicle or 50 ng ml$^{-1}$ FGF19 for 10 min, and p-Y-FXR levels were detected by IP/IB. **b** PMH expressing flag-FXR were treated with FGF19 for the indicated times, and FXR phosphorylation levels were determined by IP/IB. Relative p-Y-FXR to total FXR levels are shown at the bottom ($n = 3$). **c** PMH expressing flag-FXR were treated with vehicle or 50 ng ml$^{-1}$ FGF19, GW4064 (0.5 μM), or insulin (100 nM) for 10 min, and p-Y-FXR levels were determined by IP/IB. The p-Y-FXR level in the vehicle-treated sample was set as 1. Relative p-Y-FXR levels are shown at the bottom ($n = 3$). **d** C57BL/6 mice were fed a normal chow diet (ND) or 0.5% cholic acid (CA) chow for 3 h after 12 h of fasting and p-Y-FXR levels were detected by IP/IB. Relative p-Y-FXR levels are shown at the bottom ($n = 4$). **e** C57BL/6 mice or FXR-KO mice were fed a ND or 0.5% CA chow for 6 h or were fasted for 12 h and then treated with GW4064 (3 h) or FGF19 (30 min). The p-Y67-FXR levels in liver sections were detected by IHC. Scale bar, 50 μm. **f, g** C57BL/6 mice or FGF15-KO mice were fasted for 12 h (Fs) or fed for 6 h (Fd) after fasting. **f** p-Y67 FXR and FXR levels were measured by IB ($n = 3$). **g** p-Y67-FXR or FXR levels in liver sections were detected by IHC. Scale bar, 50 μm. All values are presented as mean ± SD. Statistical significance was measured using the **d** Mann–Whitney test, **b, c** one-way ANOVA with the Bonferroni post-test. *$P < 0.05$, **$P < 0.01$, and NS, statistically not significant

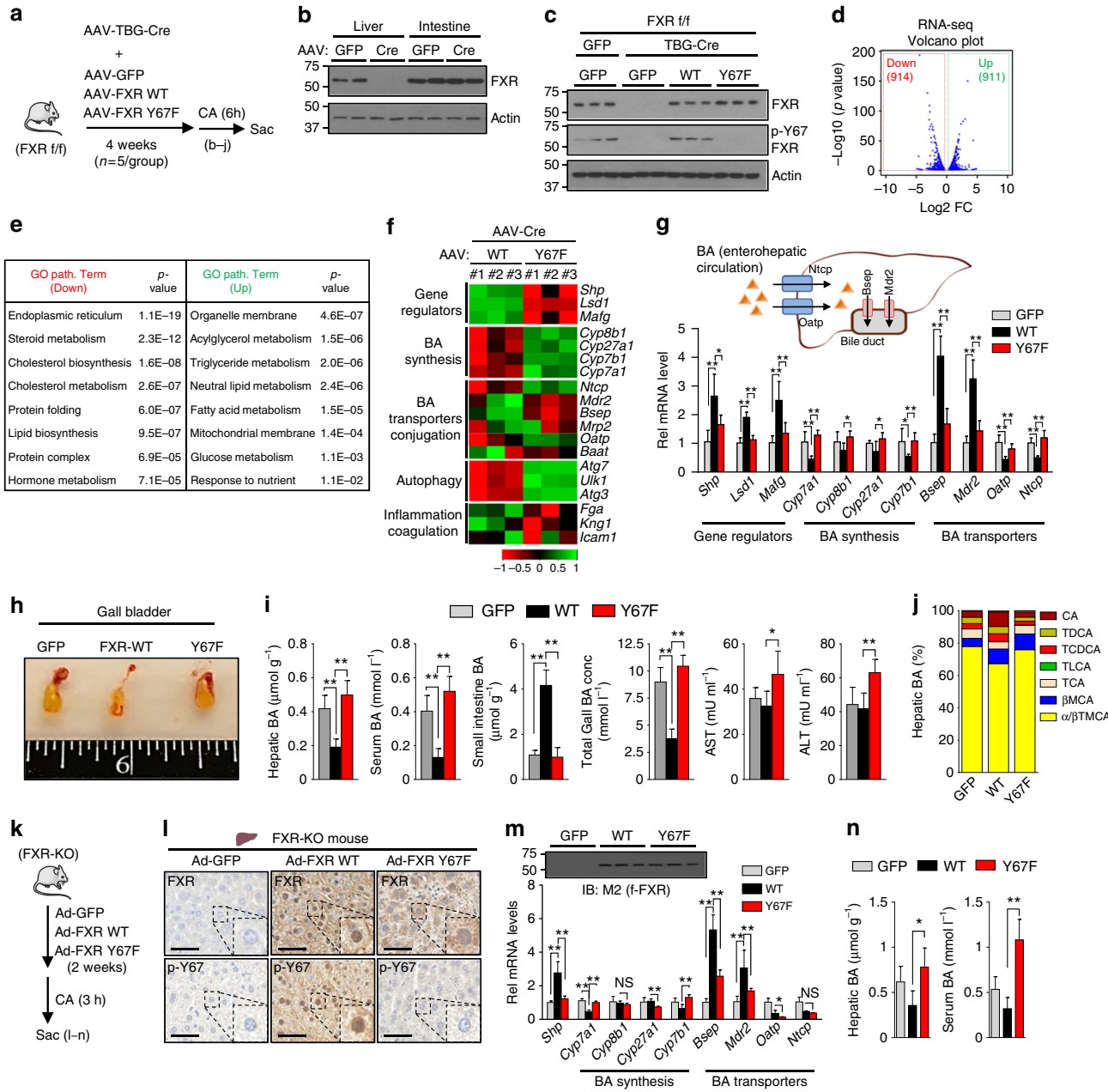

**Fig. 2** Y67F-FXR expression leads to defective BA regulation. **a–j** FXR-floxed mice were co-infected with AAV-TBG-Cre and either AAV-TBG-GFP or AAV-TBG expressing FXR-WT or Y67F-FXR for 4 weeks (5 mice/group). **a** Experimental outline. **b** FXR levels in the liver and small intestine. **c** FXR and p-Y67 FXR levels in the liver. **d–f** RNA-seq analysis: **d** Volcano plots, **e** GO analysis of genes up- or down-regulated. **f** Heat maps of changes in gene expression in mice expressing Y67F-FXR compared to mice expressing WT-FXR. **g** The mRNA levels of the indicated genes were measured by qRT-PCR. Diagram of liver BA transporters (top). **h** Images of gallbladders. **i** BA levels and serum aspartate transaminase (AST) and alanine transaminase (ALT) levels. **j** BA composition in the liver extracts. **k–n** FXR-WT, Y67F-FXR, or GFP (control) were adenovirally expressed in FXR-KO mice. **k** Experimental outline. **l** FXR and p-Y67 FXR in liver sections were detected by IHC analysis. Scale bar, 50 μm. **m** The mRNAs of the indicated genes were measured by qRT-PCR. Protein levels of FXR are shown at the top. **n** BA levels in liver and serum. **g, i, m, n** Statistical significance was measured using the one-way ANOVA with the Bonferroni post-test. *$P < 0.05$, **$P < 0.01$, and NS, statistically not significant. All values are presented as mean ± SD ($n = 5$ mice)

FXR floxed mice (Fig. 2c). To activate FXR signaling, the mice were briefly fed with CA chow before killing. Hepatic p-Y67-FXR levels were restored to the levels similar to those in the FXR-floxed mice by expression of FXR-WT, but not in mice expressing Y67F-FXR mutant (Fig. 2c).

To explore global effects of mutation of Y67 in FXR in liver, mRNA levels in mice expressing FXR-WT and Y67F-FXR were compared by RNA-seq analysis. Expression of 914 and 911 genes were significantly decreased and increased, respectively,

over 1.5-fold, in mice expressing the Y67F-FXR compared to FXR-WT (Fig. 2d). In gene ontology (GO) analysis, genes downregulated with high significance were those involved in cholesterol, BA, lipid metabolic processes, as well as, endoplasmic reticulum functions (Fig. 2e). Notably, expression of hepatic genes encoding gene repressors of BA synthesis, *Shp*, *Lsd1*, and *Mafg*, and BA exporter genes, *Bsep* and *Mdr2*, were downregulated, while BA synthetic genes, including *Cyp7a1*, and BA importer genes, *Ntcp* and *Oatp*, and direct FXR target genes

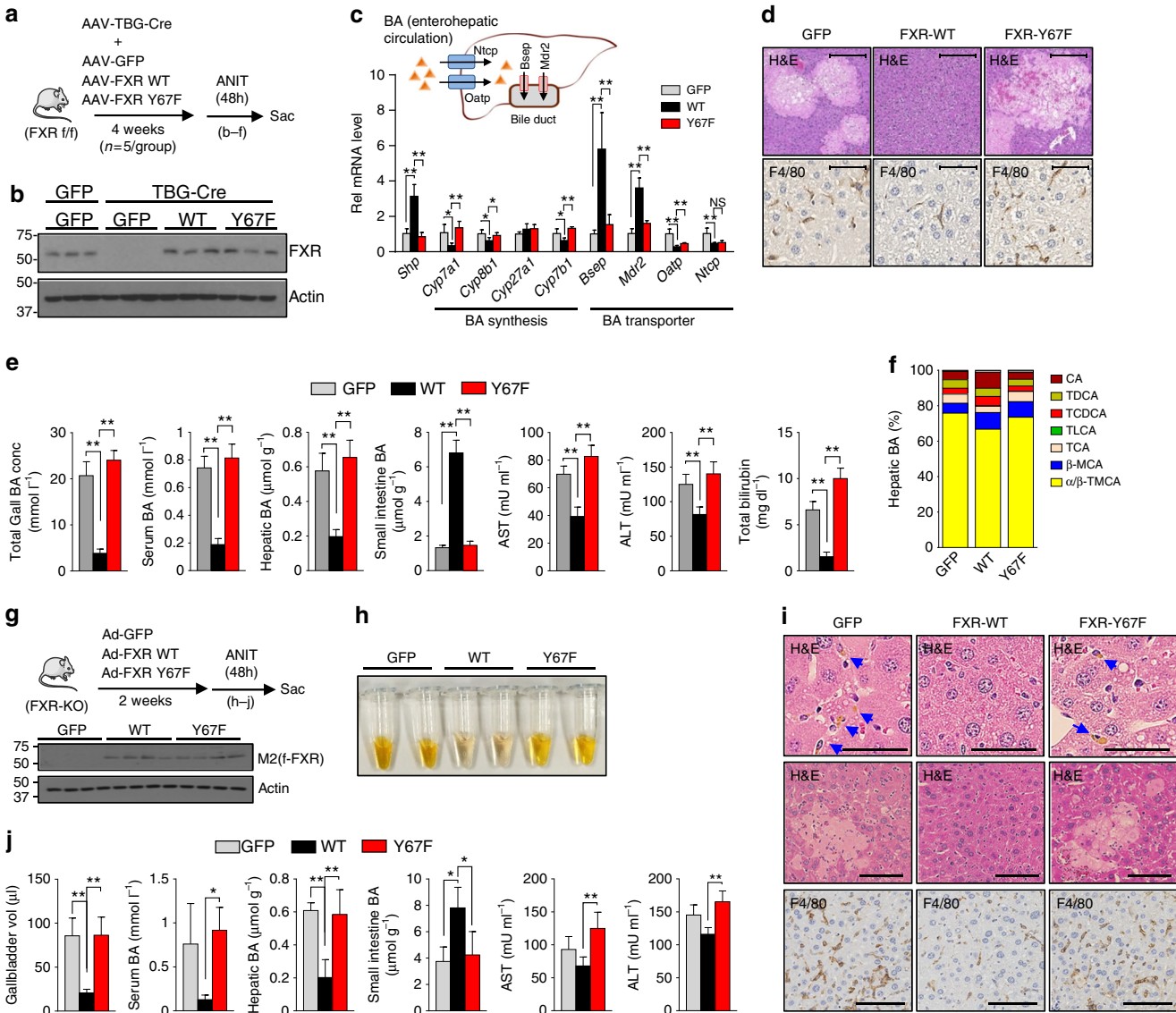

**Fig. 3** Y67F-FXR expression impairs response to induced cholestasis. **a–f** FXR-WT or Y67F-FXR was expressed in the livers of FXR floxed mice as in Fig. 2a and mice were treated with ANIT for 48 h (5 mice/group). **a** Experimental outline. **b** Levels of FXR in liver extracts detected by IB. **c** The mRNA levels of indicated genes in the liver. Diagram of liver BA transporters (top). **d** Liver sections were stained with H&E (top) and F4/80 antibody (bottom). Scale bar, 300 μm (H&E) and 50 μm (F4/80). **e** BA levels in the gallbladder, serum, liver, and intestine and serum AST, ALT and total bilirubin levels. **f** BA composition in the liver. **g–j** FXR-WT, Y67F-FXR, or GFP were adenovirally expressed in FXR-KO mice and the mice were treated with ANIT for 48 h (6 mice/group). **g** Experimental outline (top) and protein levels of FXR in liver extracts (n = 4). **h** Serum color from 2 mice in each group. **i** Liver sections were stained with H&E and F4/80 antibody. At the top, abnormally accumulated bile acids are indicated by the blue arrows. Scale bar, 50 μm (top), 100 μm (middle and F4/80). **j** The volume of the gallbladder, BA levels in serum, liver, and small intestine and serum AST and ALT levels. **c, e, j** All values are presented as mean ± SD (n = 5 or 6 mice). Statistical significance was measured using one-way ANOVA with the Bonferroni post-test. *P < 0.05, **P < 0.01, and NS statistically not significant

involved in autophagy, *Ulk1*, *Atg3*, and *Atg7*[12], were upregulated (Fig. 2f ).

Confirming the RNA-seq data, expression FXR-WT resulted in upregulation of direct FXR target genes that reduce liver BA levels, including *Shp*, *Lsd1*, *Mafg*, and *Bsep*, and downregulation of BA synthetic genes including *Cyp7a1*, and *Ntcp* and *Oatp*, but these effects were blunted by the Y67F mutation (Fig. 2g). As expected from these results, in mice expressing FXR-WT, gall bladder size was decreased (Fig. 2h), and BA levels in liver, serum, gall bladder were decreased, and hepatic BA compositions were altered (Fig. 2i, j), particularly, with decreased levels of FXR antagonists, α/βTMCA [32] (Fig. 2j, Supplementary Fig. 2), and all

these FXR-mediated effects were attenuated in mice expressing Y67F-FXR. Further, serum levels of alanine transaminase (ALT) and aspartate transaminase (AST), indicative of liver toxicity, were significantly increased in mice expressing Y67F-FXR (Fig. 2i). Similar effects on gene expression, BA levels, and liver toxicity were observed by adenoviral-mediated expression of FXR-WT or Y67F-FXR mutant in whole body FXR-KO mice (Fig. 2k–n).

The results from RNA-seq and metabolic studies utilizing AAV-mediated liver-specific expression of FXR-WT and the Y67F-FXR mutant support an in vivo role for Y67-FXR phosphorylation in regulating BA levels under normal conditions.

**Mice expressing Y67F-FXR have impaired responses to chole-static insult**. To determine if phosphorylation of FXR at Y67 is also important for adaptive responses to biliary insults, mice expressing either FXR-WT or Y67F-FXR were challenged with α-naphthyl isothiocyanate (ANIT) (Fig. 3a, b), which induces intrahepatic cholestasis by injuring biliary epithelial cells[28,33,34]. In gene expression studies of mice treated with ANIT, *Cyp7a1*, *Cyp8b1*, *Cyp7b1*, *Ntcp*, and *Oatp* were downregulated, and *Shp*, *Bsep* and *Mdr2* were upregulated in by expression of FXR-WT (Fig. 3c), which would be expected to reduce liver BA levels. These effects were nearly completely reversed by the Y67F

mutation of FXR for all the genes tested except for *Oatp* and *Ntcp*, suggesting that Y67 phosphorylation of FXR affects its function in a gene-specific manner (Fig. 3c).

Consistent with gene expression data, pathological changes in the liver induced by ANIT treatment, including necrosis, loosened hepatocyte structure, and hepatic inflammation based on macrophage infiltration were substantially improved in mice expressing FXR-WT, but not in mice expressing Y67F-FXR (Fig. 3d). In metabolic studies, expression of FXR-WT also resulted in decreased liver, serum, and gall bladder BA levels, decreased gall bladder size (Fig. 3e, Supplementary Fig. 3a, 3b),

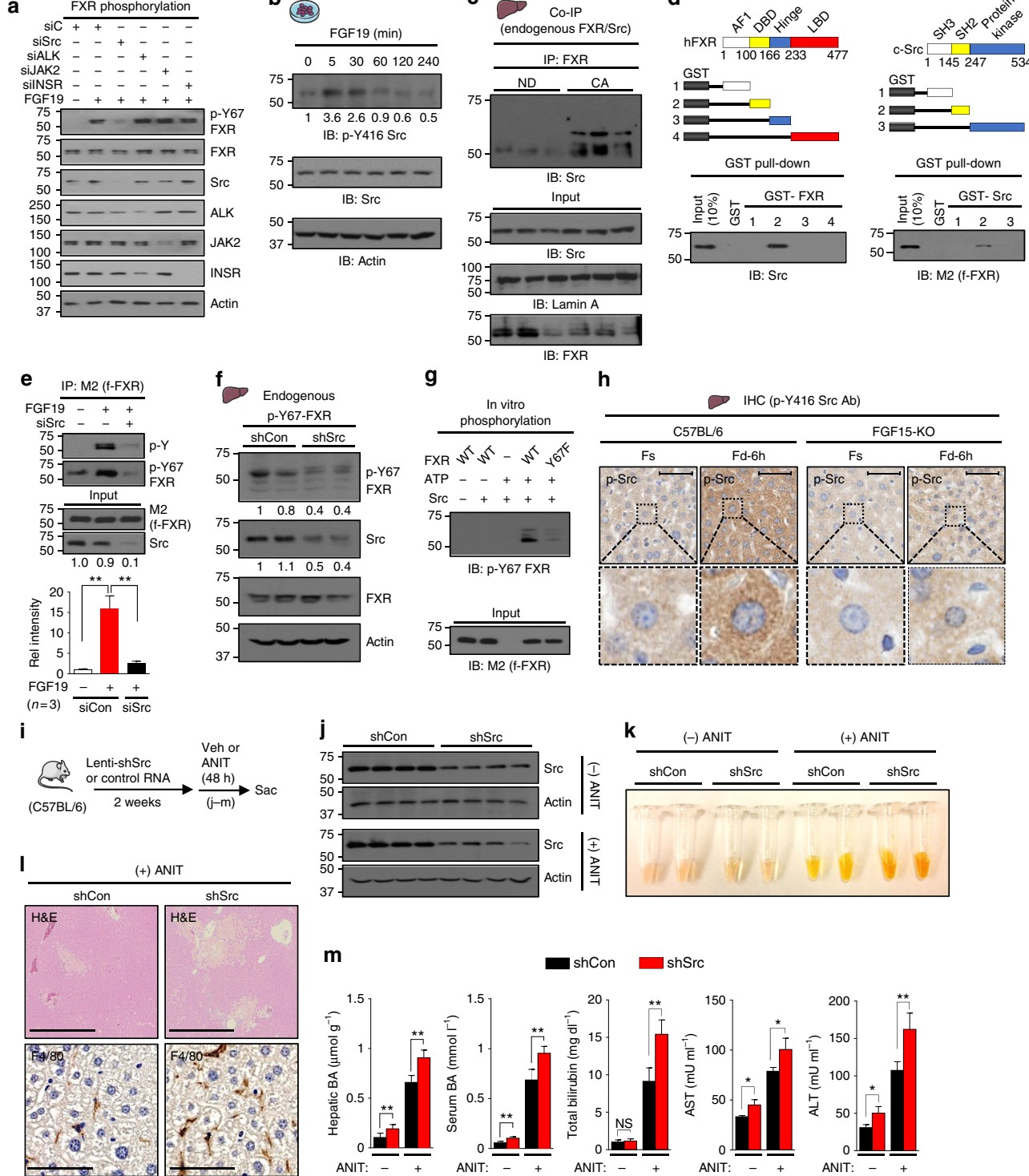

decreased levels of serum ALT/AST and bilirubin (Fig. 3e) and conjugated BAs, particularly FXR antagonists, α/βTMCA (Fig. 3f, Supplementary Fig. 3c), but all these FXR-mediated adaptive responses were attenuated by the Y67F mutation of FXR.

Similar results were observed with ANIT experiments utilizing adenoviral-mediated liver-specific expression of FXR-WT or Y67F-FXR in whole body FXR-KO mice (Fig. 3g–j, Supplementary Fig. 3d, 3e) and in C57BL/6 mice (Supplementary Fig. 3f–j), although ANIT-induced cholestatic liver pathology was more severe in FXR-KO mice compared to those in C57BL/6 mice. These results, taken together, demonstrate that FGF19-induced FXR phosphorylation at Y67 is important for adaptive responses to biliary insults that result in reduced liver BA levels and protection from cholestasis.

**Phosphorylation of Y67-FXR is mediated by c-Src kinase**. The sequence around Y67 was identified in silico as a possible motif for several Tyr kinases, including Src, ALK, and JAK2 (Supplementary Fig. 4a). In hepatocytes, FGF19 treatment increased p-Y67-FXR levels, and siRNA-mediated downregulation of Src, but not that of ALK, JAK2, or the insulin receptor, decreased the p-Y67-FXR levels (Fig. 4a), suggesting that Src kinase is likely involved in FGF19-induced phosphorylation of FXR. Supporting this idea, levels of p-Y416-Src, indicative of Src activation[35,36], were transiently elevated after FGF19 treatment (Fig. 4b), which is similar to the kinetics of Tyr-FXR phosphorylation (Fig. 1b). Further, the interaction of FXR with Src in liver extracts was dramatically induced by feeding mice CA chow (Fig. 4c), and direct interaction of FXR with the SH2 region of Src (Fig. 4d, Supplementary Fig. 4b), but not with FGFR4 (Supplementary Fig. 4c), was observed in GST-pull down analysis. Downregulation of Src with siRNA (Fig. 4e), or treatment with a Src inhibitor, dasatinib (Supplementary Fig. 4d), decreased p-Y67-FXR levels in FGF19-treated hepatocytes, and downregulation of Src in mice resulted in a marked decrease in hepatic p-Y67-FXR levels (Fig. 4f). FXR-WT, but not Y67F-FXR, was phosphorylated by addition of Src in in vitro kinase assays (Fig. 4g). These results strongly suggest that Src mediates FXR phosphorylation at Y67 in response to FGF19.

Importantly, while total Src levels were not increased after feeding (Supplementary Fig. 4e), levels of active p-Y416-Src, mainly cytoplasmic, were increased after feeding in C57BL/6 mice, but not in FGF15-KO mice (Fig. 4h). These results indicate that endogenous Src kinase in liver hepatocytes is physiologically activated by postprandial FGF15 signaling in mice.

**Responses to cholestatic insult are impaired by downregulation of Src**. Since Y67-FXR phosphorylation, which is mediated by Src, has a critical role in adaptive responses to hepatobiliary insults (Fig. 3), we next asked whether Src has a similar beneficial role and whether Src downregulation (Fig. 4i, j) led to exacerbated cholestatic liver pathology in ANIT-treated mice. ANIT treatment resulted in darker yellow serum color (Fig. 4k), increased pathological liver histology and macrophage infiltration (Fig. 4l) and increased liver BA levels and serum BA, bilirubin, and ALT/AST levels (Fig. 4m) and each of these ANIT-induced hepatobiliary toxic effects was exacerbated by downregulation of Src (Fig. 4k–m). Further, expression of Shp was decreased and that of BA synthetic genes, including Cyp7a1, was increased by Src downregulation (Supplementary Fig. 4f). These results reveal a novel function of Src in regulating BA levels and protecting against BA-related hepatotoxicity.

**Nuclear localization of the p-defective Y67F-FXR mutant is impaired**. Feeding markedly increased nuclear levels of FXR in C57BL/6 mice, but in FGF15-KO mice FXR remained mostly cytoplasmic after feeding (Fig. 1g), suggesting FGF19-mediated phosphorylation is required for nuclear localization of FXR. Thus, we examined the effect of FGF19 and the Y67F mutation FXR on its nuclear localization in hepatocytes incubated in serum-free media overnight and then treated with FGF19. In these cells, FXR was localized predominantly in the cytoplasmic fraction, and FGF19 treatment resulted in localization of FXR-WT largely in the nuclear fraction, while Y67F-FXR was mostly detected in the cytoplasm with or without FGF19 treatment (Fig. 5a). Similar effects of the Y67 mutation on nuclear localization were also observed in FGF19-treated mice (Fig. 5b) and in mice briefly fed CA chow (Supplementary Fig. 5a). Remarkably, FXR was highly concentrated in the nucleus in mice expressing FXR-WT, whereas FXR was detected mostly in the cytoplasm in mice expressing the Y67F mutant (Fig. 5c, Supplementary Fig. 5b). These results demonstrate that nuclear FXR levels are substantially decreased by the Y67F-FXR mutation, indicating that nuclear localization of FXR after FGF19 treatment or feeding is dependent on FGF19-induced phosphorylation of FXR at Y67.

**Mutation of Y67 in FXR or downregulation of Src leads to decreased interaction with RXRα and DNA binding**. We next examined effects of mutation of Y67 of FXR on its interaction with its DNA-binding partner, RXRα, and DNA binding. The interaction of exogenously expressed FXR with RXRα in extracts from FGF19-treated FXR-KO hepatocytes was reduced by the Y67F mutation (Fig. 5d, left). Consistent with the role of Src in Y67-FXR phosphorylation, downregulation of Src (Supplementary Fig. 6a) also decreased the FXR-RXRα interaction (Fig. 5d, right). In ChIP assays, occupancy of both FXR and RNA pol II at

**Fig. 4** FGF19-activated Src kinase mediates Y67-FXR phosphorylation. **a** PMH from FXR-KO were transfected with siRNA for Src, ALK, JAK2, and INSR and infected with Ad-flag-FXR, and then the cells were treated with 50 ng ml$^{-1}$ FGF19 for 10 min. Levels of indicated proteins were detected by IB. **b** PMH were treated with vehicle or FGF19 for the indicated times, and activated p-Y416-Src levels were determined by IB. **c** Mice were fed a ND or 0.5% CA chow for 3 h, and the interaction of FXR with Src was detected by CoIP ($n = 3$). **d** Fragments of FXR (left) and c-Src (right) that were fused to GST (top). Binding of Src or flag-FXR to GST-FXR or GST-Src fusion proteins, respectively, was detected by IB. **e** Flag-FXR-WT and siRNA for Src were expressed in hepatocytes which were treated with FGF19 for 10 min. p-Y-FXR and p-Y67-FXR levels were detected by IP/IB. Relative p-Y-FXR levels are shown on the bottom ($n = 3$). **f** C57BL/6 mice were infected with lenti-shRNA for Src or control RNA for 1 month, and mice were briefly fed a 0.5% CA chow, and then, p-Y67-FXR levels in liver extracts were detected by IB ($n = 2$ mice). **g** Flag-FXR-WT or Y67F-FXR was expressed in HepG2 cells, purified FXR proteins were incubated with ATP and recombinant Src (ab60884, Abcam) as indicated, and p-Y67-FXR levels were detected by IB. **h** Liver sections from C57BL/6 mice or FGF15-KO mice that had been fasted (Fs) or refed (Fd) were performed by IHC using the p-Y416-Src antibody. Scale bar, 50 μm. **i–m** C57BL/6 mice were infected with lentivirus expressing shRNA for Src for 2 weeks and treated for 48 h with ANIT or vehicle. **i** Experimental outline. **j** Protein levels of Src detected by IB. **k** Serum samples. **l** Liver sections stained with H&E or F4/80 antibody. Scale bar, 300 μm (H&E) and 100 μm (F4/80). **m** Levels of the hepatic and serum BAs and total bilirubin, AST, and ALT in serum. All values are presented as mean ± SD. Statistical significance was measured using the **e** one- or **m** two-way ANOVA with the Bonferroni post-test. *$P < 0.05$, **$P < 0.01$, and NS, statistically not significant

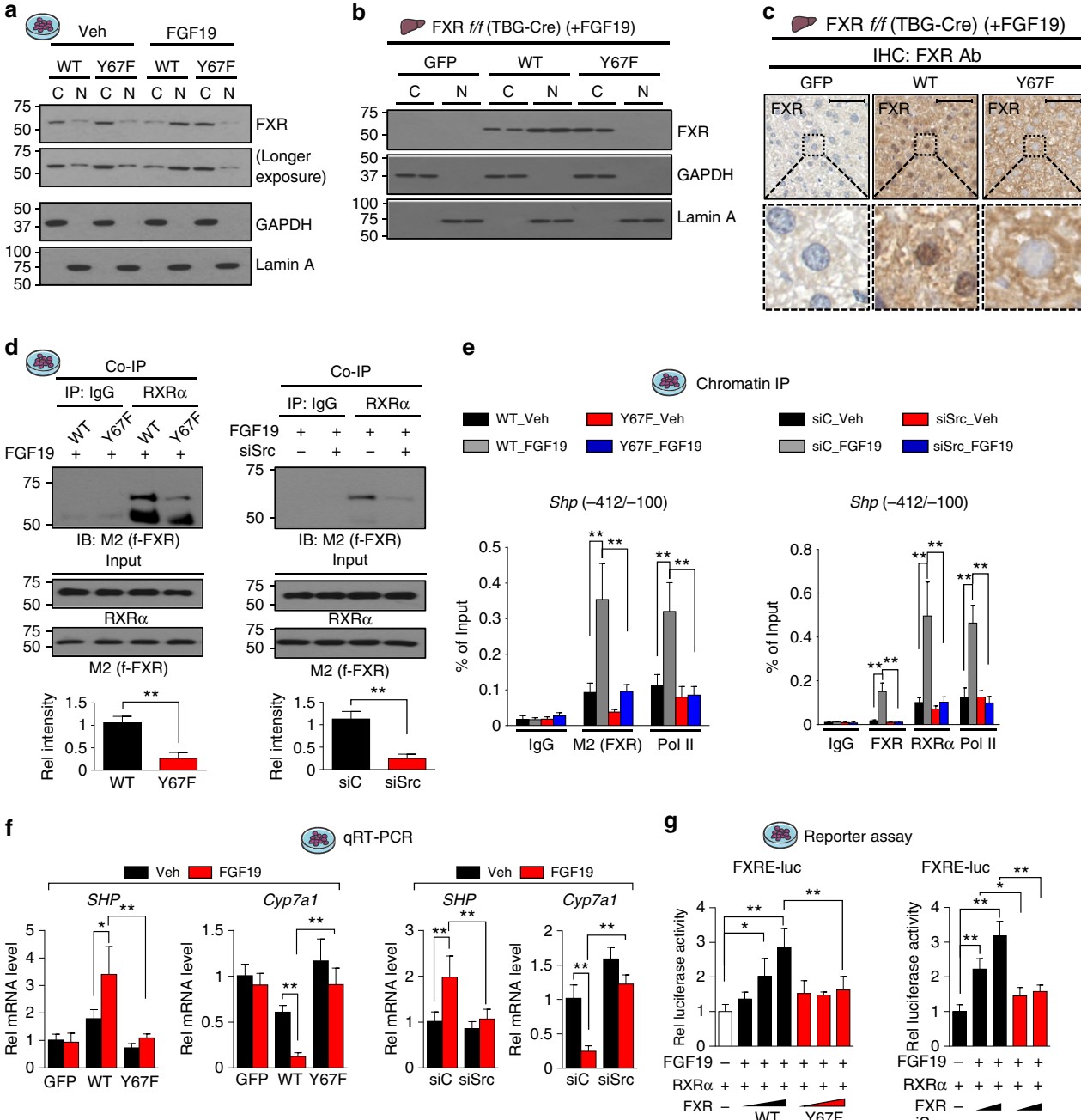

**Fig. 5** Phosphorylation of FXR at Y67 is important for its function. **a** PMH expressing FXR-WT or Y67F-FXR were treated with vehicle or FGF19 for 10 min and nuclear (N) and cytoplasmic (C) extracts were isolated. Levels of FXR in N or C fractions was measured by IB. **b**, **c** FXR-WT or Y67F-FXR was expressed in FXR-floxed mice infected with AAV-TBG-Cre as in Fig. 2a, and the mice were treated with FGF19 for 30 min. **b** Levels of FXR in nuclear and cytoplasmic liver extracts were determined by IB ($n = 2$). **c** FXR levels in liver sections were detected by IHC. Scale bar, 50 μm. **d** PMH from FXR-KO mice were infected with Ad-flag FXR WT or Y67F-FXR (left) or transfected with siRNA for Src (right) and then, treated with FGF19 for 30 min. FXR in anti-RXRα immunoprecipitates was detected by IB. Band intensities are shown on the bottom with the WT and siC values set to 1 ($n = 3$). **e**, **f** Flag-FXR-WT or Y67F-FXR was adenovirally expressed in PMH from FXR-KO mice (left), or endogenous Src was downregulated by siRNA in PMH from C57BL/6 mice (right). Then, cells were treated with vehicle or FGF19 for 30 min. **e** Occupancy of flag-FXR and RNA pol II was detected by ChIP, and **f** mRNA levels of *Shp* and *Cyp7a1* were measured by qRT-PCR ($n = 3$–5). **g** PMH were transfected with plasmids or siRNA as indicated and treated with vehicle or FGF19 for 3 h and the luciferase activities were measured and normalized to β-galactosidase activities ($n = 4$). All values are presented as mean ± SD. Statistical significance was measured using the **d** Mann–Whitney test, **g** one- or **e**, **f** two-way ANOVA with the Bonferroni post-test. *$P < 0.05$, **$P < 0.01$

the promoter of the FXR target, *Shp*, was impaired by the Y67F-FXR mutation (Fig. 5e, left) or by downregulation of Src (Fig. 5e, right).

Consistent with these results, FGF19 treatment led to increased mRNA levels of *Shp* and decreased levels of *Cyp7a1* in cells

expressing FXR-WT, but these effects were blunted with Y67F-FXR (Fig. 5f, left) or after downregulation of Src (Fig. 5f, right). Further, expression of FXR-WT increased transactivation of FXRE-Luc in a dose-dependent manner in FGF19-treated cells, but these effects were not observed with Y67F-FXR (Fig. 5g, left)

or after downregulation of Src (Fig. 5g, right). Notably, expression of p-mimic Y67E-FXR in hepatocytes from FXR-KO mice led to increased basal expression of *Shp* with decreased expression of *Cyp7a1* (Supplementary Fig. 6b), and increased FXR transactivation of FXRE-Luc reporter (Supplementary Fig. 6c) without FGF19 treatment. These results, together, indicate that the

FGF19-induced Src phosphorylation of FXR at Y67 is important for increased transcriptional activity of FXR, as well as its nuclear localization.

**FGF19-induced phosphorylation of FXR by Src is dependent on Shp2.** The non-receptor tyrosine phosphatase, Shp2, acts as a

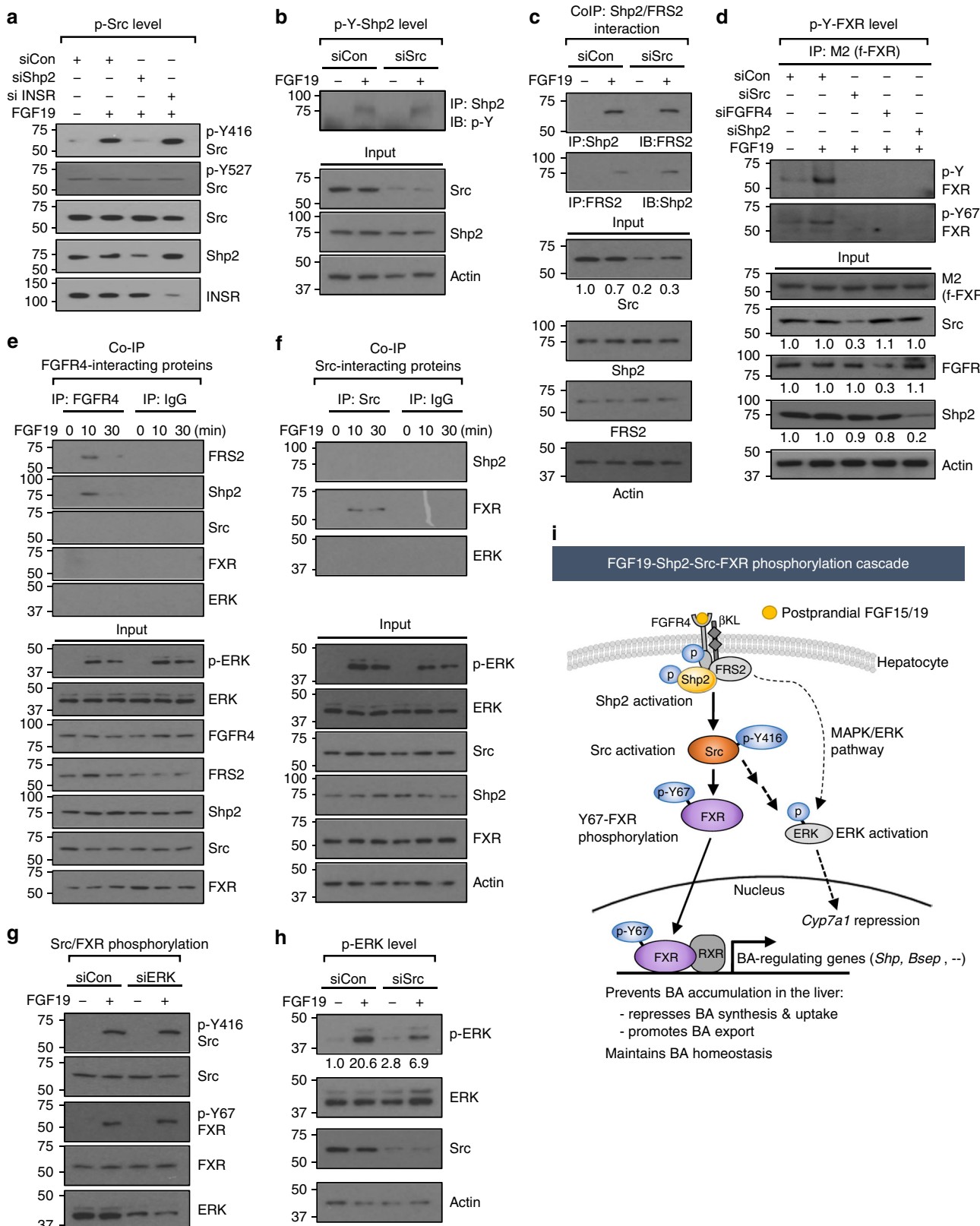

key component of FGF19 signaling and coordinates hepatic regulation of BA-induced FXR and FGF19-FGFR4 signaling to suppress BA synthesis[37]. Since we observed that FXR is phosphorylated at Y67 by activated Src in response to postprandial FGF15/19 signaling (Fig. 4), we next examined whether activation of Src is dependent on Shp2.

In hepatocytes, treatment with FGF19 resulted in increased levels of p-Y416-Src, an activated Src marker[35,36], but not that of p-Y527-Src, a repressed Src marker[35,36], and the increases in the level of activated p-Y416-Src after FGF19 treatment were blocked by downregulation of Shp2, but not by downregulation of the insulin receptor (Fig. 6a). Further, downregulation of Src did not affect the activated p-Y-Shp2 level, an indicator of Shp2 activation[37] (Fig. 6b). These results suggest that Y416-Src phosphorylation is downstream of Shp2 and independent of insulin signaling. In CoIP studies, FGF19 treatment increased Shp2 interaction with FRS2 as reported[37] and downregulation of Src did not alter this interaction (Fig. 6c). Notably, downregulation of Shp2 or FGFR4, as well as Src, decreased the FGF19-induced p-Y-FXR and p-Y67-FXR levels (Fig. 6d). These results, together, indicate that FGF19-activation of Shp2 is upstream of Src and that Shp2 is required for the Src-mediated FXR phosphorylation.

Since phosphorylation of FXR and Src are downstream of Shp2, we next asked whether FXR and Src, like Shp2 and FRS2[37], are present in a complex with FGFR4. In hepatocyte extracts, FGF19 stimulation transiently induced the interaction of FGFR4 with FRS2 or Shp2, but not with Src, FXR, or ERK (Fig. 6e). Further, FGF19 treatment induced the interaction of Src with FXR, but not with Shp2 or ERK (Fig. 6f ). These results suggest that Src and FXR are not likely part of the FGFR4/FRS2/Shp2 complex at the plasma membrane.

**FGF19-activated Src contributes to activation of ERK independently of the MAPK signaling pathway.** Previous studies have shown that pharmacological inhibition or siRNA-mediated downregulation of ERK diminished FGF15/19-mediated repression of Cyp7a1 in human hepatocytes and mouse livers[38,39]. Since ERK was shown to act downstream of FGF19 signaling-activated Shp2[37], we further tested whether the ERK is in the pathway leading to Src activation for FXR phosphorylation. In hepatocytes, downregulation of ERK did not block the increased levels of activated p-Y416-Src and p-Y67-FXR after FGF19 treatment (Fig. 6g), while downregulation of Src partially decreased p-ERK levels (Fig. 6h). Further, downregulation of ERK did not affect the FGF19-induced interaction of FXR with Src (Supplementary Fig. 7a). These results suggest that ERK is downstream of Src and can also be activated by Src upon FGF19 signaling.

Our findings, together, support a model (Fig. 6i) in which binding of postprandial FGF19 to the receptor complex, FGFR4 and βKL, in hepatocytes results in activation of FGF19-FGFR4-FRS2-Shp2 signaling pathway. This signaling leads to activation of Src. FGF19 signaling-activated Src physically interacts with and phosphorylates FXR at Y67, and this Y67-FXR phosphorylation is important for nuclear localization of FXR and its transcriptional regulation of BA levels to maintain BA homeostasis and to prevent BA accumulation in the liver. Further, FGF19-activated Src also activates ERK independently of the MAPK/ERK signaling pathway[5,40], which results in transcriptional repression of BA synthesis[38,39].

**Hepatic FGF19-Src-FXR signaling is likely defective in primary biliary cirrhosis (PBC) patients.** Aberrantly elevated hepatic BA levels lead to cholestatic liver injury, which can further develop into fatal diseases, such as fibrosis, cirrhosis, and cancer[6,7]. Serum and liver FGF19 levels are highly elevated in PBC patients and correlate with the disease severity, suggesting that FGF19 signaling is impaired in these patients[41]. To determine whether the hepatic FGF19-Src-FXR signaling pathway is impaired in PBC patients, we examined hepatic expression of FXR, SRC, and the FGF19 receptor complex, FGFR4 and βKL in 15 normal subjects and PBC patients.

The mRNA levels of FXR and SRC were substantially decreased in PBC patients compared to normal subjects (Fig. 7a). Hepatic FGFR4 mRNA levels were highly elevated, but intriguingly, mRNA levels of its obligate co-receptor βKL, were dramatically downregulated in these patients (Fig. 7a). This dramatic decrease in βKL mRNA suggests that FGF19 signaling is defective in PBC patients.

Protein levels of p-Y416-Src, p-Y67-FXR, FXR, SRC, and a known indicator of FGF19 signaling, p-ERK[1,22,42], were markedly reduced in the PBC patients (Fig. 7b, c). While both the total and phosphorylated forms of Src and FXR were decreased, the ratio of phosphorylated forms to total protein was decreased (Fig. 7d), suggestive of impaired FGF19 signaling in livers of PBC patients. These findings indicate that the hepatic FGF19-Src-FXR phosphorylation pathway is likely defective in PBC patients.

## Discussion

In this study, we demonstrate that FXR is a direct physiological target of postprandial FGF15/19-induced phosphorylation mediated by Src, and that this FXR phosphorylation is critical for its transcriptional functions that maintain BA homeostasis under normal conditions and mediate adaptive responses upon cholestatic insults. Further, a critical role for the FXR-FGF19 endocrine axis in regulating BA levels has been previously established[17–19],

**Fig. 6** FGFR4-Shp2 are required for Y67-FXR phosphorylation by Src. **a** PMH were transfected with siRNA as indicated, and 48 h later, cells were treated with FGF19 for 10 min, and p-Y416-Src, p-Y527-Src, Src, Shp2 and insulin receptor (INSR) levels were detected by IB. **b** PMH were transfected with siRNA for Src as indicated and treated with FGF19 for 10 min, and p-Y-Shp2 levels were detected by IP/IB. Protein levels in input samples were detected by IB. **c** PMH were transfected with siRNA for Src as indicated and treated with FGF19 for 10 min, and FRS2 or Shp2 in anti-Shp2 or anti-FRS2 immunoprecipitates, respectively, were detected by IB. Protein levels in input samples were detected by IB. **d** PMH were transfected with siRNA as indicated and infected with Ad-flag-FXR, and then, 48 h later, the cells were treated with FGF19 for 10 min. p-Y-FXR and p-Y67-FXR levels were detected by IP/IB. Protein levels in input samples were detected by IB, and relative intensities of the bands, determined using Image J, are plotted below the blots. **e** CoIP: PMH were treated with FGF19 for the indicated times, and FRS2, Shp2, Src, FXR, and ERK in anti-FGFR4 immunoprecipitates were detected by IB. **f** CoIP: hepatocytes were treated with FGF19 for 10 or 30 min, and Shp2, FXR, and ERK were detected by IB in anti-Src immunoprecipitates. Protein levels in input samples are shown below. **g** PMH were transfected with siRNA for ERK and treated with FGF19 for 10 min, and levels of indicated proteins were detected by IB. **h** PMH were transfected with siRNA for Src and treated with FGF19 for 10 min, and levels of Src, p-ERK, and ERK were detected by IB. **i** Model: an FGF19-FGFR4-Shp2 signaling cascade leads to activation of Src and Src-mediated phosphorylation of FXR at Y67, which is important for nuclear localization of FXR and transcriptional regulation of BA levels by FXR. FGF19-activated Src also activates ERK independently of the MAPK/ERK signaling pathway, which leads to transcriptional repression of Cyp7a1[38,39]

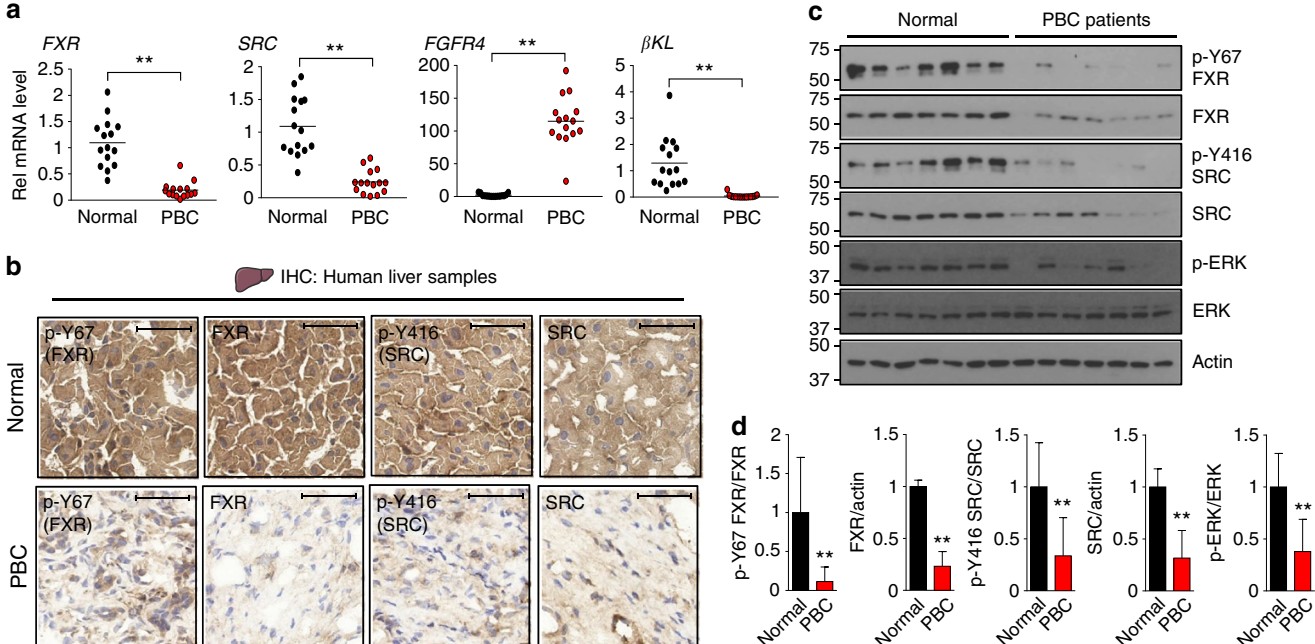

**Fig. 7** The FGF19-Src-FXR axis is likely defective in PBC patients. **a** Hepatic mRNA levels of the indicated genes in PBC patients were measured by qRT-PCR ($n = 15$). **b** Liver sections from normal and PBC patients were performed by IHC using the p-Y67-FXR, FXR, p-Y416-Src, and Src antibodies. Scale bar, 50 μm. **c** Hepatic levels of indicated proteins in randomly selected 7 normal subjects and PBC patients were measured by IB. **d** Expression levels of the indicated proteins are shown with the normal values set to 1. All values are presented as mean ± SD. **a**, **d** Statistical significance was measured using the Student $t$-test. **$P < 0.01$

and we now show that activation of the membrane FGF19 receptor is transmitted to the nucleus for transcriptional control of BA levels through a pathway involving an FGF19-Src-FXR phosphorylation cascade.

RNA-seq and metabolic studies (Fig. 2) strongly support an in vivo role for the FGF19-induced Y67-FXR phosphorylation in BA regulation. In mice expressing the Y67F-FXR mutant, direct FXR targets that reduce hepatic BA levels, including *Shp* and *Bsep*, were downregulated, while BA synthetic genes, including *Cyp7a1*, were upregulated. BA levels were elevated, and particularly, hepatic levels of FXR antagonistic BAs were significantly increased. Further, mice expressing the Y67F-FXR mutant had exacerbated cholestatic symptoms upon exposure to ANIT (Fig. 3). Notably, this hepatobiliary damage in mice expressing the Y67F-FXR mutant were similar to those in FGF15-KO, FGFR4-KO, and βKL-KO mice[20,23,24], suggesting that Y67-FXR phosphorylation is a key component in the hepatic FGF19-FGFR4 pathway.

Mechanistic studies suggested that the Y67F phosphorylation is important for nuclear localization, DNA binding, and transcriptional activity of FXR (Fig. 5). In contrast, the phospho-mimic Y67E-FXR mutant exhibited increased transcriptional activity without FGF19 treatment (Supplementary Fig. 6b, 6c), which provide strong evidence that the effects of the Y67F-FXR mutant result specifically from blocking phosphorylation at Y67. In gene expression studies, we observed striking effects of Y67-FXR mutation on expression of *Bsep*. Intriguingly, a recent study has shown that mutations in the human FXR gene cause neonatal intrahepatic cholestasis, which is associated with undetectable *Bsep* expression as a consequence of loss of FXR function[43].

The cytoplasmic tyrosine phosphatase Shp2 has been identified as an indispensable key regulator of *Cyp7a1* expression and hepatic BA synthesis[37]. Remarkably, liver-specific ablation of Shp2, which acts immediately downstream of the FGF19 receptor, inhibited FGF19 signaling and attenuated FXR induction of *Shp*,

resulting in elevated BA levels and hepatobiliary damage in mice. However, Shp2 did not form a complex with FXR[37], so the signaling pathway linking Shp2 to FXR was unidentified. The present study identifies non-receptor tyrosine kinase, Src, as a downstream target of Shp2, linking the FGF19-FGFR4-Shp2 signaling to the phosphorylation of FXR (Fig. 6i).

Src plays an important role in cell growth, differentiation, and the immune response[35], but, a role for Src in metabolic regulation has not been reported. In our studies, downregulation of Src blocked FGF19-mediated phosphorylation of FXR, suggesting that BA regulation would be affected which was confirmed by the increased liver and serum BA levels and exacerbation of liver injury and hepatic inflammation in Src-downregulated mice. Although the other Src family kinases could be functionally redundant with Src, the nearly complete block of the FGF19-induced increase in FXR phosphorylation by downregulation of Src, together with our findings from biochemical analyses including direct FXR interaction with Src, suggests that this is not the case. Src may regulate BA levels through phosphorylation of other nuclear receptors in the liver, such as HNF-4, which occurs in colon cancer[44], or LRH-1, but our results suggest that Src phosphorylation of FXR plays a key role in BA regulation.

It has been shown that FGF15/19 signaling activates ERK, which contributes to repression of *Cyp7a1*[1,5,20,22,38,42]. While FGF19 signaling leads to the activation of ERK via the MAPK signaling pathway[5,40], our findings in signaling cross-talk studies in hepatocytes suggest that FGF19-activated Src also independently activates ERK (Fig. 6i). In addition, FGF19 signaling leads to activation of PKCζ and subsequently, phosphorylation of SHP, which is important for its nuclear localization and *Cyp7a1* repression[45]. We observed that FGF19 treatment increases activated p-PKCζ levels and downregulation of PKCζ resulted in decreased p-Y416-Src and p-Y67-FXR levels, while it did not affect the p-Y-Shp2 level (Supplementary Fig. 7b). These results suggest that FGF19-mediated activation of Shp2 is independent of

PKCζ and that FGF19-Shp2 and FGF19-PKCζ signaling pathways likely converge on activation of Src. Further studies will be required to elucidate how FGF19-activated PKCζ positively influences the Src-mediated phosphorylation of FXR.

BA levels in enterohepatic tissues temporally fluctuate after feeding. BA concentrations in the intestine rapidly increase after a meal, and the majority of BAs are returned to the liver and then, stored in the gall bladder. In humans, BA concentrations in the enterohepatic portal vein increase between 15 and 60 min after a meal, which activates hepatic FXR in the early fed-state, while FXR-mediated induction of intestinal FGF19 results in FGF19 peak levels later, about 3 h, after a meal[21,31], so that FGF19 acts on liver hepatocytes in the late fed-state. Thus, postprandial FGF19-induced FXR phosphorylation is likely important for sustaining transcriptional functions of FXR to regulate BA levels in the late fed-state as hepatic BA levels decrease. Interestingly, treatment with CDCA, which activates both FXR and the membrane BA receptor, also transiently increased Y67-FXR phosphorylation (Supplementary Fig. 8). The mechanism by which activation of membrane BA signaling increases the FXR phosphorylation and the relationship to FGF19-mediated phosphorylation will require further studies.

One of the intriguing findings in this study is that the hepatic FGF19-Src-FXR pathway is likely defective in PBC patients. Hepatic mRNA levels of *FXR* and *SRC* were substantially lower, and protein levels of p-Y67-FXR and activated p-Y416-SRC were dramatically decreased in these patients. FGF19 levels are highly elevated in PBC patients, suggesting defective FGF19 signaling in these patients[41]. Consistent with this observation, activated p-ERK levels, a known downstream kinase of FGF15/19 signaling[1,22,42], were also significantly decreased about 60% in the PBC patients. Further, while hepatic expression of *FGFR4* was highly elevated, expression of the essential co-receptor *βKL*, was dramatically decreased, which strongly suggests that FGF19 signaling is defective in PBC patients. A recent study has shown that *βKL* is a direct target of FXR in hepatocytes and increases FGFR4 protein stability[16].

The FXR-FGF19 gut–liver endocrine pathway has received great attention because treatment with FXR agonists, such as, a synthetic BA analog, obeticholic acid (OCA), and a gut-specific FXR agonist, fexaramine, and FGF19 analogs, has beneficial effects on amelioration of hepatobiliary damage, as well as on the metabolic syndrome[17,19,46]. In particular, OCA, which was recently FDA-approved for treatment of liver fibrosis and PBC, has high affinity for FXR and increases intestinal expression of FGF19[19]. It will be intriguing to see whether these OCA-mediated effects are dependent on FGF19-induced phosphorylation of FXR by Src, which as shown in the present study, is important for the transcriptional function of FXR in reducing liver BA levels and maintaining BA homeostasis. Disruption of BA homeostasis is strongly associated with the development of cholestatic liver disease, which can progress to cirrhosis, fibrosis, and cancer. The hepatic FGF19-Src-FXR signaling axis, thus, provides a potential drug target and its components may be novel biomarkers for PBC and other enterohepatic diseases related to BA dysregulation.

## Methods

**Animal experiments**. Eight week-old male C57BL/6 or FXR-KO mice were fasted for 6 h before killing to avoid metabolic fluctuations. For AAV-mediated FXR expression experiments, eight week-old FXR floxed male mice were injected via the tail vein with 100 μl of a mixture of purified AAV serotype 8 (AAV8) containing a liver-specific TBG promoter driving Cre recombinase and either AAV8-TBG-GFP, AAV8-TBG-FXR WT, or AAV8-TBG-Y67F-FXR at $5 \times 10^{10}$ virus particles per mouse, and four weeks later, mice were fed 0.5% CA chow for 6 h to activate FXR signaling or treated by gavage with ANIT for 48 h. Adenovirus expressing GFP, FXR WT, or Y67F mutant ($2.5–5.0 \times 10^8$ active viral particles in 100 μl saline) was injected via the tail vein, and two weeks later, serum and tissues were collected.

Lentivirus expressing control shRNA or shRNA (VectorBuilder, Inc) for Src ($0.5–1.0 \times 10^9$ IU per ml in 100 μl saline) was injected via the tail vein and after 1 month, mice were treated by gavage with vehicle (olive oil) or 75 mg kg$^{-1}$ ANIT for 48 h[28,33,34]. Investigators were not blinded to the group allocation during the experiment or when assessing the outcome. No formal randomization was used. All animal use and viral protocols were approved by the University of Illinois, Urbana-Champaign Institutional Animal Care and Biosafety Committees.

**Reagents**. Antibodies for FXR (1:5000, sc-13063), Lamin A (1:5000, sc-20680), ALK (1:3000, sc-398791), JAK2 (1:3000, sc-390539), INSR (1:10,000, sc-57344), GAPDH (1:5000, sc-166574), RNA Pol II (sc-9001), FGFR4 (1:5000, sc-136988), Shp2 (1:3000, sc-280), FRS2 (1:3000, sc-17841) and RXRα (1:5000, sc-553) were purchased from Santa Cruz Biotechnology and for p-Tyr (1:5000, 8954), p-ERK (1:3000, 9101), ERK (1:3000, 4695), PKCζ (1:5000, 9368), β-actin (1:10,000, 4970), Src (1:3000, 2108), p-Y527 Src (1:3000, 2105) and p-Y416 Src (1:3000, 2101) from Cell Signaling. M2 antibody (1:3000, F3165) and M2 agarose (A2220) were purchased from Sigma, Inc. F4/80 (AP10243PU-M) was obtained from Acris Antibodies. Antibodies for phospho-Tyr-67-specific FXR were produced commercially (1:10,000, Abmart, Inc.) and validated in C57BL/6 and FXR-KO mice by IHC in this study (Fig. 1e). BAs, TCDCA, TDCA, TCA, and CA, were purchased from Sigma Inc, α/β-TMCA and TLCA from Santa Cruz Biotech, and βMCA from TRC, Inc. GW4064 was obtained from Tocris bioscience. Src inhibitor (dasatinib monohydrate) was purchased from Selleckchem. Expression plasmid for FGFR4 was purchased from VectorBuilder. ON-TARGET*plus* mouse siRNAs for Src (L-040877), ALK (J-040104), JAK2 (J-040118), INSR (J-043748), FGFR4 (L-045345), and Shp2 (L-041173) were purchased from GE Healthcare Dharmacon, Inc. siRNA for ERK1/2 (6560) was purchased from Cell Signaling. HepG2 cells were obtained from ATCC (HB-8065).

**Adenoviral and lentiviral vector constructions**. The phosphorylation-defective flag-FXR mutant was constructed by site-directed mutagenesis (Stratagene, Inc.) and confirmed by sequencing. Flag-FXR in this manuscript refers to 3 × Flag-human FXR[26,27]. Purified lentiviruses for sh-control and shRNA for Src were purchased from VectorBuilder.

**RNA-Seq**. The mRNA fragment library was prepared using the RNeasy mini prep kit (Qiagen) according to the manufacturer's instructions. cDNAs of the RNA in the library was sequenced using Illumina HiSeq2000 (Illumina, San Diego, CA) to produce paired-end 100-bp reads, summarized as "left" and "right" reads. One library of reads per biological sample was examined for sequencing errors prior to mapping to genome and transcriptome features. Quality of sequencing was examined using FastQC 15 with acceptable scores about 30, a Qphred quality value which is the negative logarithmic transformation of the estimated probability of error. Sequencing alignment was performed by STAR ver 2.5.0a. Gene ontology analysis was performed using DAVID (david.abcc.ncifcrf.gov).

**BA composition analysis**. Liver samples were analyzed with the 5500 QTRAP LC/MS/MS system (Sciex, Framingham, MA) in the Metabolomics Center, University of Illinois at Urbana-Champaign. The LC separation was performed by HPLC (1200 series HPLC system, Agilent Technologies, Santa Clara, CA) using an Agilent Eclipse Plus XDB-C18 column (4.6 × 100 mm, 3.5 μm) with mobile phase A (10 mM ammonia formate) and mobile phase B (methanol). Mass spectra were acquired under negative electrospray ionization. Multiple reaction monitoring was used for quantitation.

**In-cell and in vivo FXR phosphorylation assays**. PMH were treated with FGF19 and tyrosine phosphatase inhibitors for 2 min. For in vivo studies, livers were collected from mice fasted overnight. Flag-FXR or endogenous FXR was immunoprecipitated from freshly prepared cell or liver extracts. Phosphorylation of FXR at Y67 was detected using the p-Y67 FXR-specific antibody by IB.

**Nuclear localization study**. Liver tissue was minced and resuspended in hypotonic buffer (10 mM Hepes, 1.5 mM MgCl$_2$, 10 mM KCl, 0.2% NP40, 1 mM EDTA, and 5% sucrose) and cells were lysed by homogenization[28,45]. Nuclei were pelleted by cushion buffer (10 mM Tris-HCl, pH 7.5, 15 mM NaCl, 60 mM KCl, 1 mM EDTA, and 10% sucrose). The nuclear pellet and cytoplasmic supernatant were collected after centrifugation with 5000 rpm for 3 min. For the isolation of nuclear and cytoplasmic fractions of PMH, NE-PER Nuclear and Cytoplasmic Extraction Reagents purchased from Thermo-Fisher Scientific. For nuclear localization studies in cells, PMH isolated from FXR-KO mice were infected with Ad-FXR WT or Y67F-FXR and then, the cells were incubated with serum-free media overnight and treated with FGF19 for 30 min. Cytoplasmic GAPDH and nuclear lamin A were detected for assessing the quality of cellular fractionation.

**Liver histology and toxicity**. For IHC studies, paraffin-embedded liver sections were incubated with the indicated antibody overnight at 4 °C and detected with the rabbit-specific HRP/DAB Detection IHC Kit (Abcam). The sections were stained with hematoxylin and eosin (H&E) and imaged with a NanoZoomer Scanner

(Hamamatsu). Serum ALT/AST levels, BA levels and total bilirubin levels were measured using colorimetric analysis kits from Sigma, Trinity Biotech, and Bio-vision, respectively.

**Tandem mass spectrometry analysis.** Flag-human FXR was adenovirally expressed in PMH isolated from FXR-KO mice and 48 h later, the cells were treated with 5 μM MG132 for 4 h to inhibit proteasomal degradation of FXR[26] and then treated with 50 ng ml$^{-1}$ FGF19 for 10 min. Flag-FXR was purified using M2 agarose (Sigma, Inc) and subjected to LC–MS/MS proteomic analysis[26–28].

**PMH isolation and luciferase reporter.** PMHs were isolated by collagenase (0.8 mg ml$^{-1}$, Sigma, Inc) perfusion through the portal vein of mice anesthetized with isoflurane. Hepatocytes were filtered through a cell strainer (100 μm nylon, BD), washed with M199 medium, resuspended in M199 medium, centrifuged through 45% Percoll (Sigma, Inc.), and cultured in M199 medium containing 10% fetal bovine serum[15,16,28]. In luciferase reporter assays, PMH from FXR-KO mice, to avoid confounding effects due to endogenous FXR, were transfected with expression plasmids for FXR-WT or Y67F-FXR. PMH were transfected with 250 ng reporter plasmids, 300 ng of β-galactosidase plasmid, 50 ng of plasmids for RXRα, and 20–60 ng of expression plasmids for flag-FXR WT or Y67F or transfected with siRNAs, and 48 h later, the cells were treated with vehicle or 50 ng ml$^{-1}$ FGF19 for 3 h. Luciferase activities were normalized to β-galactosidase activities.

**ChIP assays and cell signaling studies in PMH.** For ChIP assay, PMH were infected with Ad-flag-FXR WT or Y67F-FXR or transfected with siRNAs for control or Src, and 48 h later, cells were treated with FGF19 for 30 min, and then, cells were washed twice with PBS, then incubated with 1% formaldehyde for 10 min at 37 °C. Glycine was added to 125 mM for 5 min at room temperature. Chromatin solutions in sonication buffer (50 mM Tris-HCl, pH 8.0, 2 mM EDTA, and 1% SDS) with protease inhibitors were sonicated four times with 10 s intervals using a QSonica XL-2000 instrument at power output setting 8. Then, chromatin sample was precleared and chromatin was immunoprecipitated using 1–1.5 μg of antibody or IgG as control. The immune complexes were collected by incubation with a Protein G–Sepharose slurry (Invitrogen) containing salmon-sperm DNA for 1 h, washing with 0.1% SDS, 1% Triton X-100, 2 mM EDTA, 20 mM Tris-HCl, pH 8.0, three times containing successively 150 mM NaCl, 500 mM NaCl, or 0.25 M LiCl, and then eluted and incubated overnight at 65 °C to reverse the crosslinking. DNA was isolated for qPCR.[12,15,26,27] The qPCR primer sequences are as follows: Mouse *Shp*, 5′-CAGTGAGAACCCTGGTCTT-3′ (forward) and 5′-CTGGCCAAACAACCTTGAC-3′ (reverse). For cell signaling studies, PMH were transfected with siRNAs for 48 h, treated with FGF19 for 10 min, and IB analyses were done.

**Quantification of mRNA.** RNA was isolated from liver and quantified by qRT-PCR, normalized to 36B4 mRNA. The qRT-PCR primer sequences used are as follows: Mouse *36B4*, 5′-CGACATCACAGAGCAGGC-3′ (forward) and 5′-CACC GAGGCAACAGTTGG-3′ (reverse); Mouse *SHP*, 5′-TCTGCAGGTCGTCCGAC-TAT-3′ (forward) and 5′-CAGGCAGTGGCTGTGAGAT-3′ (reverse); Mouse *Lsd1*, 5′-AGCAGCCTGTTTCCCAGACA-3′ (forward) and 5′-TGCAATGTGCGA TTCCTGAT-3′ (reverse); Mouse *Mafg*, 5′-GACCCCCAATAAAGGAAACAA-3′ (forward) and 5′-TCAACTCTCGCACCGAGAT-3′ (reverse); Mouse *Cyp7a1*, 5′-CATCTCAAGCAAACACCATTCC-3′ (forward) and 5′-TCACTTCTTCAGAG GCTGGTTTC-3′ (reverse); Mouse *Cyp8b1*, 5′-GAATCTAACCAGGCCATGCT-3′ (forward) and 5′-AGGAGCTGGCACCTAGACT-3′ (reverse); Mouse *Cyp27a1*, 5′-GACAACCTCCTTTGGGACTTAC-3′ (forward) and 5′-GTGGTCTCTTATTGG GTACTTGC-3′ (reverse); Mouse *Cyp7b1*, 5′-GACGATCCTGAAATAGGAGCAC A-3′ (forward) and 5′-AATGGTGTTTGCTAGAGAGGCC-3′ (reverse); Mouse *Bsep*, 5′-CAATGTTCAGTTCCTCCGTTCA-3′ (forward) and 5′-TTTGGTGTTGT CCCCGTGCTTG-3′ (reverse); Mouse *Mdr2*, 5′-CACAGAACACAGCCAACCT-3′ (forward) and 5′-AACAACCGATAACAGCAGAAGT−3′ (reverse); Mouse *Oatp*, 5′-CCTGGAGCAGCAATATGGAAA-3′ (forward) and 5′-CCAAGGCATACTGG AGGCAA-3′ (reverse); Mouse *Ntcp*, 5′-TACCTCCTCCCTGATGCCTTTC-3′ (forward) and 5′-TGCGTCTGCAGCTTGGATTTA-3′ (reverse); Human *36B4*, 5′-TTGGCTACCCAACTGTTGCA-3′ (forward) and 5′-CACAAAGGCAGATGGA TCAGC-3′ (reverse); Human *FXR*, 5′-GCAGCCTGAAGAGTGGTACTCTC-3′ (forward) and 5′-CATTCAGCCAACATTCCCATCTC-3′ (reverse); Human *SRC*, 5′-GGACAGTGGCGGATTCTACATC-3′ (forward) and 5′-AGCTGCTGCAGG CTGTTGA-3′ (reverse); Human *FGFR4*, 5′-AGTATCTGGAGTCCCGGAA-3′ (forward) and 5′-CCAGCCCAAAGTCAGCAAT-3′ (reverse); and Human *βKL*, 5′-ACATTTACATCACCGCCAG-3′ (forward) and 5′-AGATTTCTCTTCAGC-CAGTT-3′ (reverse).

**CoIP and GST pull-down assays.** CoIP studies using whole cell extracts from liver or PMH were prepared in CoIP buffer (50 mM Tris, pH 8.0, 150 mM NaCl, 2 mM EDTA, 0.3% NP40, 10% glycerol) and incubated with 2 μg of antibody or control IgG as indicated for 60 min, and 35 μl of 25% Protein G–Sepharose slurry was added. After 2 h, agarose beads were washed with the CoIP buffer and bound proteins were detected by IB.[14,27,28] For GST pull-down assay, DNA fragments encoding different regions of FXR and Src were inserted into the pGEX4T-1 vector.

Bacterially expressed and affinity purified GST-fusion proteins were incubated with the reciprocal proteins that were synthesized by TNT (Promega, Inc), and bound proteins were detected by IB.

**PBC patient study.** Liver specimens from 15 unidentifiable healthy individuals or from 15 PBC patients were obtained from the Liver Tissue Cell Distribution System that operates under a contract from the National Institutes of Health and ethical approval was not required.

**Statistical analysis.** GraphPad Prism 6 was used for data analysis. Statistical significance was determined by Student's two-tailed *t*-test, Mann–Whitney test or one- or two-way ANOVA with Bonferroni post-test for single or multiple comparisons as appropriate. Whenever relevant, the assumptions of normality were verified using the Shapiro–Wilk test, Kolmogorov–Smirnov test or the D'agostino–Pearson omnibus test. *P*-values < 0.05 were considered as statistically significant.

**Data availability.** The data that support the findings of this study are available from the corresponding author upon reasonable request. The RNA-seq data are deposited in the Gene Expression Omnibus (GEO) database with the Accession Number GSE113707.

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

## Acknowledgements

We thank Johan Auwerx and Kristina Schoonjans at Ecole Polytechnique Fédérale de Lausanne, Switzerland, for kindly providing FXR-floxed mice. We also thank the Liver Tissue Cell Distribution System, Minneapolis, University of Minnesota (NIH Contract # HHSN276201200017C) for providing human liver specimens of PBC patients. Petri dish, mouse, and liver images in the figures are unmodified images provided by Servier Medical Art by Servier, licensed under a Creative Commons Attribution 3.0 Unported License [https://creativecommons.org/licenses/by/3.0/legalcode]. This study was supported by an American Heart Association Postdoctoral Fellowship to S.B. (17POST33410223), an American Heart Association Scientist Development Award (16SDG27570006) to Y.-C.K., and by R01 grants from the National Institutes of Health (DK062777 and DK095842) and an Innovative Basic Science grant from the American Diabetes Association (1-16-IBS-156) to J.K.K.

## Author contributions

S.B., D.-H.K., D.R., and J.K.K. designed research; S.B., D.-H.K., D.R., Y.-C.K., and H.S. performed experiments; B.Ko., G.G., and H.E.X. provide key materials for the study, P.Y. performed proteomic analyses; S.B., D.-H.K., D.R., Y.-C.K., H.S., B.Ke., and J.K.K. analyzed data; and S.B., D.-H.K., D.R., B.Ke., and J.K.K. wrote the paper.

## Additional information

**Competing interests:** The authors declare no competing interests.

