## [Peer Review File · Nature Communications]

Reviewers' comments:

Reviewer #1 (Remarks to the Author):

This work by Byun and colleagues explores the mechanism by which the ileal-derived endocrine hormone FGF19 transmits its signal to hepatocytes to regulate FXR activation and bile acid signaling. FXR phosphorylation sites were initially identified using LC-MS/MS of a FXR-FLAG construct re-expressed in primary hepatocytes from FXR KO mice treated with FGF19. The authors then performed mutagenesis of three identified phospho-sites and focused on Y67 based on functional data and in silico phospho-analysis of FXR. To identify the kinase responsible for phosphorylating FXR at residue 67, the authors utilize siRNA studies and observed significantly decreased phospho-Y67-FXR levels when Src is knocked down in mouse hepatocytes and validate interaction with Co-IP and GST pull down studies as well as in vitro phosphorylation studies. The authors then utilize in vitro and in vivo studies to demonstrate that phosphorylation of Y67 is needed for nuclear localization of FXR in hepatocytes/liver. Overall, this manuscript is well written and the data are presented in a clear and logical manner.

Comment:

The authors tested phosphorylation of FXR at Y67 in response to fasting and feeding in WT and FGF15-KO mice in vivo (Fig 1F) as well testing the effect of Y67F-FXR mutation on nuclear translocation in mouse hepatocytes in response to FGF19 (Fig 5A). They also test the effect of acutely expressing the phospho-FXR mutant in livers in vivo on nuclear translocation. However, they do not test the effect of expressing this mutant in vivo on nuclear translocation of FXR in vivo in response to rFGF19 administration. This experiment would provide in vivo support of the proposed model.

Reviewer #2 (Remarks to the Author):

Byun and colleagues' manuscript describes that after postprandial increase of fibroblast growth factor-19/15 there is activation of its receptor FGFR4 aided by its co-receptor beta-KL resulting in the phosphorylation of the nuclear receptor FXR (farnesoid-X-receptor) at tyrosine-67 regulating in this manner its transcriptional activity, maintaining the correct homeostasis of bile acids by downregulating their excess production, avoiding in this manner cholangitis and hepatobiliary damage. They present extensive experimental evidences showing that FXR is phosphorylated by c-Src, which is activated upon FGFR4 activation by its ligand FGF19. They also present compelling evidence that a Tyr67Phe FKR mutant, when expressed in liver or c-Src downregulation with siRNA, alters the response of this organ to bile acids overload and increase the severity of drug-induced cholestasis. Further, they have preliminary data (15 cases) showing that in primary biliary cholangitis patients there is low expression of the FGFR4 co-receptor beta-KL, altering therefore the signaling pathway implicated in bile acids homeostasis.

Major points:

1. The mechanism by which FGFR4/beta-KL complex transmits the signal to attain c-Src activation, however, is not identified in all its steps. They implicate the tyrosine-phosphatase Shp2 that upon interaction with the adaptor protein FRS2 (FGFR substrate 2) is coupled to the FGFR/beta-KL complex. But the role of this tyrosine-phosphatase remains unexplained. Phosphorylation of c-Src at its regulatory Tyr527 residue by the C-terminal Src kinase (Csk), or its homologue kinase (Chk), acts as a negative regulator of its activity, and subsequent dephosphorylation of this residue activates again the kinase (see for review Roskoski (2005) Biochem. Biophys. Res. Commun.324: 1155-1164). Could Shp2 dephosphorylate phospho-Tyr527 in c-Src activating in this manner the kinase? This should be tested.
2. They propose FGFR4-dependent activation of protein kinase Czeta (PKCzeta), which phosphorylates the anomalous nuclear receptor/transcription factor SHP, which appear to act

rather as a co-receptor. Does PKCzeta phosphorylates Shp2 too? Several other PKC isoforms phosphorylate Shp2 although its phosphatase activity is not affected upon phorbol ester treatment (Strack et al. 2002 *Biochemistry* 41: 603-608). Downregulation of PKCzeta with siRNA, although totally prevents FXR phosphorylation at Tyr67 does not totally block c-Src activation (Fig. S6), showing that PKCzeta is not an absolute necessary component of the signaling cascade.

3. Although they indicate that c-Src independently leads to ERK activation (page 14), Fig. 6h shows that FGF19-mediated ERK phosphorylation is partially operative after strong downregulation of c-Src expression using siRNA. FGFR4 is expected to activate the MAPK independently of c-Src. Correct? Please explain if this is a possible scenario.

4. In Fig. 4a although the expression of c-Src after siRNA treatment is undetectable, phospho-Y67 FXR is present. Could FGFR4 directly phosphorylate FXR? Please discuss this point.

5. Hemoglobin contaminated some serum samples in Fig. 4k preventing the observation of its real color. This panel should be replaced avoiding hemolysis during sample preparation.

Other points:

1. Residue Tyr-Y67. This label is redundant, as it uses the three letters and one letter amino acid code at the same time. Please use Y67 or Tyr67 not both together.

2. In the abstract and page 16, please write "phospho-defective" or "phospho-mimic" instead of "phosphor-defective" or "phosphor-mimic".

3. The abbreviations ALT, alanine transaminase, and AST, aspartate transaminase, should be indicated in the text.

4. In page 17 there is a phrase that requires correction: "... from biochemical analyses including showing that FXR directly...".

5. In page 18, the phrase "It will be interesting to see whether AMPK-induced FXR phosphorylation affects Y67-FXR phosphorylation" is confusing, as AMPK is a serine/threonine kinase. It should be clarified that it refers to c-Src mediated phosphorylation of Tyr67. Correct?

6. qRT-PCR is some time written q-RTPCR. Please unify.

7. References list. The Doi number is some time followed by [doi] or [pii] (e.g. refs. 17, 18, 27, 33-35, 37, 38, 42). Please delete. The Doi number is duplicated in ref. 41. Capitals is used in the title of some references (e.g. refs. 19, 21). Please unify.

8. Legend of Fig. 1. The abbreviation ND, normal chow diet, and CA chow (containing 0.5% cholic acid) should appear a couple of lines above from where they appear now.

9. Please define in the legend the abbreviations LBD and DBD in Fig. S4b.

10. Please define the abbreviation CDCA, chenodeoxy cholic acid in the legend of Fig. S7.

Reviewer #1 (Remarks to the Author):

This work by Byun and colleagues explores the mechanism by which the ileal-derived endocrine hormone FGF19 transmits its signal to hepatocytes to regulate FXR activation and bile acid signaling. FXR phosphorylation sites were initially identified using LC-MS/MS of a FXR-FLAG construct re-expressed in primary hepatocytes from FXR KO mice treated with FGF19. The authors then performed mutagenesis of three identified phospho-sites and focused on Y67 based on functional data and *in silico* phospho-analysis of FXR. To identify the kinase responsible for phosphorylating FXR at residue 67, the authors utilize siRNA studies and observed significantly decreased phospho-Y67-FXR levels when *Src* is knocked down in mouse hepatocytes and validate interaction with Co-IP and GST pull down studies as well as *in vitro* phosphorylation studies. The authors then utilize *in vitro* and *in vivo* studies to demonstrate that phosphorylation of Y67 is needed for nuclear localization of FXR in hepatocytes/liver. Overall, this manuscript is well written and the data are presented in a clear and logical manner.

Comment:

The authors tested phosphorylation of FXR at Y67 in response to fasting and feeding in WT and FGF15-KO mice *in vivo* (Fig 1F) as well testing the effect of Y67F-FXR mutation on nuclear translocation in mouse hepatocytes in response to FGF19 (Fig 5A). They also test the effect of acutely expressing the phospho-FXR mutant in livers *in vivo* on nuclear translocation. However, they do not test the effect of expressing this mutant *in vivo* on nuclear translocation of FXR *in vivo* in response to rFGF19 administration. This experiment would provide *in vivo* support of the proposed model.

Response: We thank the reviewer for raising this important issue. To address this issue, we examined the levels of FXR-WT and phospho-defective Y67F-FXR expressed by AAV infection in nuclear and cytoplasmic liver extracts from mice treated with recombinant FGF19. As expected from the original studies with FGF15-KO mice showing that nuclear localization of FXR and FXR phosphorylated at Y67 after feeding is dependent on FGF15 (Fig. 1f, g) and that treatment with FGF19 in mice increased nuclear levels of endogenous FXR phosphorylated at Y67 (Fig. 1e), in revision studies, we observed that FGF19 treatment resulted in predominantly nuclear localization of FXR-WT, while Y67F-FXR remained in the cytoplasm (New Fig. 5b, c and presented below). Together with the study showing that FXR phosphorylated at Y67 is predominantly detected in the nucleus after treatment with cholic acid, GW4046, or FGF19 (Fig. 1e) or after feeding (Fig. 1g), these results provide strong evidence that FGF19-induced phosphorylation of FXR at Y67 mediates nuclear localization of FXR in liver hepatocytes.

Fig. 5. Src phosphorylation of FXR at Y67 is important for nuclear localization of FXR. (b, c) FXR-WT or Y67F-FXR was expressed in FXR-floxed mice infected with AAV-TBG-Cre and the mice were treated with FGF19 for 30 min. (b) Levels of FXR in nuclear and cytoplasmic liver extracts were determined by IB (n=2). (c) FXR levels in liver sections were detected by IHC.

Reviewer #2 (Remarks to the Author):

Byun and colleagues' manuscript describes that after postprandial increase of fibroblast growth factor-19/15 there is activation of its receptor FGFR4 aided by its co-receptor beta-KL resulting in the phosphorylation of the nuclear receptor FXR (farnesoid-X-receptor) at tyrosine-67 regulating in this manner its transcriptional activity, maintaining the correct homeostasis of bile acids by downregulating their excess production, avoiding in this manner cholangitis and hepatobiliary damage. They present extensive experimental evidences showing that FXR is phosphorylated by c-Src, which is activated upon FGFR4 activation by its ligand FGF19. They also present compelling evidence that a Tyr67Phe FKR mutant, when expressed in liver or c-Src downregulation with siRNA, alters the response of this organ to bile acids overload and increase the severity of drug-induced cholestasis. Further, they have preliminary data (15 cases) showing that in primary biliary cholangitis patients there is low expression of the FGFR4 co-receptor beta-KL, altering therefore the signaling pathway implicated in bile acids homeostasis.

Major points:

1. The mechanism by which FGFR4/beta-KL complex transmits the signal to attain c-Src activation, however, is not identified in all its steps. They implicate the tyrosine-phosphatase Shp2 that upon interaction with the adaptor protein FRS2 (FGFR substrate 2) is coupled to the FGFR/beta-KL complex. But the role of this tyrosine-phosphatase remains unexplained. Phosphorylation of c-Src at its regulatory Tyr527 residue by the C-terminal Src kinase (Csk), or its homologue kinase (Chk), acts as a negative regulator of its activity, and subsequent dephosphorylation of this residue activates again the kinase (see for review Roskoski (2005) *Biochem. Biophys. Res. Commun.* 324: 1155-1164). Could Shp2 dephosphorylate phospho-Tyr527 in c-Src activating in this manner the kinase? This should be tested.

Response: We thank the reviewer for raising this issue. To better understand how Src is activated upon FGF19 signaling, we examined the effects of FGF19 treatment and downregulation of Shp2 or INSR on levels of p-Y416-Src (an activation mark) and p-Y527-Src (a repression mark). Treatment with FGF19 resulted in marked increases in activated p-Y416-Src levels in a Shp2-dependent manner in primary mouse hepatocytes. In contrast, FGF19 treatment did not markedly change levels of p-Y527-Src and downregulation of Shp2 did not lead to marked changes in p-Y527-Src levels (New Fig. 6a and presented below). These results suggest that FGF19-mediated signaling pathway leads to activation of Src through phosphorylation at Y416-Src, but it does not involve the signaling pathway leading to dephosphorylation of Y527-Src.

Fig. 6. (a) PMH were transfected with siRNA as indicated, and 48 h later, cells were treated with FGF19 for 10 min, and p-Y416-Src, p-Y527-Src, Src, Shp2 and insulin receptor (INSR) levels were detected by IB.

2. They propose *FGFR4*-dependent activation of protein kinase *Czeta* (*PKCzeta*), which phosphorylates the anomalous nuclear receptor/transcription factor *SHP*, which appear to act rather as a co-receptor. Does *PKCzeta* phosphorylates *Shp2* too? Several other *PKC* isoforms phosphorylate *Shp2* although its phosphatase activity is not affected upon phorbol ester treatment (Strack et al. 2002 *Biochemistry* 41: 603-608).

Response: In response to the reviewer's comment, we examined the effect of downregulation of *PKCζ* on FGF19-mediated phosphorylation of several relevant proteins (new data presented below). FGF19 treatment resulted in increased levels of p-Y-Shp2, a marker of *Shp2* activation (Li et al., *Cell Metabolism*, 2014), as well as p-Y416-Src, p-Y67-FXR and p-T55-SHP. Downregulation of *PKCζ* blocked phosphorylation of Src, SHP and FXR, but did not affect the p-Y-Shp2 levels (New Fig. S7b and presented below). These results suggest that FGF19-mediated activation of *Shp2* is likely independent of *PKCζ* activation and that in parallel FGF19 independently activates *PKCζ*, and these two signaling pathways likely converge on Src for FXR phosphorylation.

Fig. S7. (b) PMH from C57BL/6 mice were transfected with siRNA for *PKCζ* as indicated and the cells were treated with FGF19 for 10 min. Protein levels of *PKCζ*, p-Y67 FXR, FXR, p-SHP (T55), SHP, p-Src (Y416), Src, p-Y Shp2 and Shp2 were detected by IB.

3. Although they indicate that *c-Src* independently leads to ERK activation (page 14), Fig. 6h shows that FGF19-mediated ERK phosphorylation is partially operative after strong downregulation of *c-Src* expression using siRNA. *FGFR4* is expected to activate the MAPK independently of *c-Src*. Correct? Please explain if this is a possible scenario.

Response: We appreciate the reviewer for pointing out this important issue. As stated, *FGFR4* activates ERK likely via the MAPK/ERK pathway (Hylemon, J. *Lipid Research*, 2009; Carter et al., *Trends in cell biology*. 2015) (Chiang, *Comprehensive Physiology*, 2013), but our data suggest that ERK can also be activated by Src (Fig. 6g, h) independently of the MAPK pathway. In the revised manuscript, we have modified our model (Fig. 6i) by including the MAPK signaling pathway and have modified the discussion to note the partial effect of Src downregulation on ERK activation and the likely independent activation of ERK through the *FGFR4*-activated MAPK pathway.

4. In Fig. 4a although the expression of *c-Src* after siRNA treatment is undetectable, phospho-Y67 FXR is present. Could *FGFR4* directly phosphorylate FXR? Please discuss this point.

Response: While it is difficult to completely exclude a possibility that *FGFR4* directly phosphorylates FXR, we think it is unlikely. While Src is not detectable in the blot in the siSrc sample, it is likely that the downregulation is not 100% and the level of p-Y67-FXR is very

substantially reduced. In CoIP assays, FGF19 treatment transiently increased the interaction of FGFR4 with FRS2 or Shp2, but not with Src, FXR, or ERK (Fig. 6e). To further address this point, we performed additional GST pull down assays in which FXR interaction with FGFR4 was not detected (New Fig. S4c and presented below) while interaction with Src was readily detected (Fig. 4d). These results, suggest that FGFR4 does not directly phosphorylate FXR in response to FGF19 signaling.

Fig. S4. (c) Fragments of FXR were fused to GST (top). FGFR4 protein was synthesized in vitro using TNT system (Promega, Inc) and binding of FGFR4 to GST-FXR fusion proteins was detected by IB.

5. Hemoglobin contaminated some serum samples in Fig. 4k preventing the observation of its real color. This panel should be replaced avoiding hemolysis during sample preparation.

Response: We agree with the reviewer’s comment. In the revision studies, we observed that serum color is darkly yellow in ANIT-treated mice and Src downregulation resulted in more darkly yellow serum color, which is consistent with increased serum bilirubin levels and exacerbated cholestatic liver pathology. New data are now presented in Fig. 4k in the revision manuscript and presented below.

Fig. 4. (k) C57BL/6 mice were infected with lentivirus expressing shRNA for Src for 2 weeks and treated for 48 h with ANIT or vehicle. Serum colors were determined (n= 2 mice).

Other points:

1. Residue Tyr-Y67. This label is redundant, as it uses the three letters and one letter amino acid code at the same time. Please use Y67 or Tyr67 not both together.

Response: Correction has been made as suggested.

2. In the abstract and page 16, please write “phospho-defective” or “phospho-mimic” instead of “phosphor-defective” or “phosphor-mimic”.

Response: Correction has been made as suggested.

3. The abbreviations ALT, alanine transaminase, and AST, aspartate transaminase, should be indicated in the text.

Response: The abbreviations for ALT and AST are indicated in the text as suggested.

4. In page 17 there is a phrase that requires correction: "... from biochemical analyses including showing that FXR directly...".

Response: Correction has been made as suggested.

5. In page 18, the phrase "It will be interesting to see whether AMPK-induced FXR phosphorylation affects Y67-FXR phosphorylation" is confusing, as AMPK is a serine/threonine kinase. It should be clarified that it refers to c-Src mediated phosphorylation of Tyr67. Correct?

Response: We agree with the reviewer it may create some confusions. We thus this sentence was deleted in the revised manuscript.

6. qRT-PCR is some time written q-RTPCR. Please unify.

Response: Correction has been made as suggested.

7. References list. The Doi number is some time followed by [doi] or [pii] (e.g. refs. 17, 18, 27, 33-35, 37, 38, 42). Please delete. The Doi number is duplicated in ref. 41. Capitals is used in the title of some references (e.g. refs. 19, 21). Please unify.

Response: Correction has been made as suggested.

8. Legend of Fig. 1. The abbreviation ND, normal chow diet, and CA chow (containing 0.5% cholic acid) should appear a couple of lines above from where they appear now.

Response: Correction has been made as suggested.

9. Please define in the legend the abbreviations LBD and DBD in Fig. S4b.

Response: The abbreviations of LBD and DBD are indicated in the legend of Fig. S4b.

10. Please define the abbreviation CDCA, chenodeoxy cholic acid in the legend of Fig. S7.

Response: The abbreviation of CDCA is indicated in the legend of Fig. S8.

We thank the reviewers for their constructive comments. We have addressed all of the concerns raised by the reviewers and the changes have strengthened the revision manuscript. We hope that it will now be acceptable for publication.

REVIEWERS' COMMENTS:

Reviewer #1 (Remarks to the Author):

The authors have successfully addressed my concern and strengthened their manuscript.

Reviewer #2 (Remarks to the Author):

The authors addressed adequately all the points of concern.

Response to reviewers

Reviewer #1 (Remarks to the Author):

Comment:

The authors have successfully addressed my concern and strengthened their manuscript.

Response: *We thank the reviewer*

Reviewer #2 (Remarks to the Author):

Comment:

The authors addressed adequately all the points of concern.

Response: *We thank the reviewer*